# Bipartite expander Hopfield networks as self-decoding high-capacity error correcting codes

**Rishidev Chaudhuri**
Center for Neuroscience,
Departments of Mathematics and
Neurobiology, Physiology and Behavior,
University of California, Davis,
Davis, CA 95616
rchaudhuri@ucdavis.edu

**Ila Fiete**
Brain and Cognitive Sciences,
Massachusetts Institute of Technology,
Cambridge, MA 02139
fiete@mit.edu

## Abstract

Neural network models of memory and error correction famously include the Hopfield network, which can directly store—and error-correct through its dynamics—arbitrary $N$-bit patterns, but only for $\sim N$ such patterns. On the other end of the spectrum, Shannon's coding theory established that it is possible to represent exponentially many states ($\sim e^N$) using $N$ symbols in such a way that an optimal decoder could correct all noise upto a threshold. We prove that it is possible to construct an associative content-addressable network that combines the properties of strong error correcting codes and Hopfield networks: it simultaneously possesses exponentially many stable states, these states are robust enough, with large enough basins of attraction that they can be correctly recovered despite errors in a finite fraction of all nodes, and the errors are intrinsically corrected by the network's own dynamics. The network is a two-layer Boltzmann machine with simple neural dynamics, low dynamic-range (binary) pairwise synaptic connections, and sparse expander graph connectivity. Thus, quasi-random sparse structures—characteristic of important error-correcting codes—may provide for high-performance computation in artificial neural networks and the brain.

## 1 Introduction

Neural systems must be able to recover stored states from partial or noisy cues (pattern completion or cleanup), the definition of an associative memory. If the memory state can be addressed by its content, it is furthermore called content-addressable. The classic framework for neural associative content-addressable (ACA) memory is the conceptually powerful Hopfield network [1–3].

Here, we are motivated by both the Hopfield network and by strong error-correcting codes (ECCs). ECCs are simultaneously compact, allowing the specification of exponentially many coding states using codewords that grow only linearly in size; and well-separated or robust, permitting the correction of relatively large errors by an appropriate decoder. Specifically, ECCs can use strings of $N$ bits to encode exponentially-many messages ($\sim 2^{\alpha N}, 0 < \alpha < 1$), and to retrieve them in the presence of noise that corrupts a finite fraction of bits [4, 5].

However, in ECCs, the decoder is external to the system and its costs (in time, space, and computational complexity) are not taken into account when determining the capacity of the code. By contrast, the simple neural dynamics of Hopfield networks permits them to cleanup or decode their own states, though only for a small number of such states. Here we pose (and answer) the question of whether it is possible to build simple, Hopfield-like neural networks that can represent exponentially many

well-separated states using linearly many neurons, and perform error-correction or decoding on these states in response to errors on a finite fraction of all neurons.

Very generally, in any representational system there is a tradeoff between capacity and noise robustness[4]: a robust system must have redundancy to recover from noise, but the redundancy comes at the price of fewer representational states. Conversely, a system with a large number of coding states leaves less room for redundancy or surrounding basins for robustness. The entire state space of a neural network with $N$ binary nodes is $2^N$, thus the goal of achieving exponentially many stable coding states or codewords—the same scaling as the total number of states—simultaneously with large basin sizes around each codeword for robust correction of errors that occur at a finite rate at every neuron (for a total number of errors linear in network size) is highly non-trivial.

We show that a bipartite neural network and its stochastic equivalent, a Boltzmann machine[6]— when equipped with random expander-graph connectivity between layers and clustered inhibition in the hidden layer—achieves the desiderata of strong error-correcting codes with built-in decoding, meaning that the network exhibits exponential capacity, robustness to large errors, and self decoding or clean-up of these errors. By forging connections between the theory of low-density parity-check (LDPC) codes on expander graphs [7], and robust representation in neural networks, this construction leads to ACA networks with an unprecedented combination of robustness and number of states.

## 2   Results

For this work, we define a Hopfield network to have $N$ binary neurons, symmetric (undirected) weights, and asynchronous updates of a single randomly selected neuron at each discrete time-step:

$$\boldsymbol{x}_i^{t+1} = \begin{cases} 1 & \text{if } \sum_j W_{ij}\boldsymbol{x}_j^t + b_i > 0 \\ 0 & \text{if} \sum_j W_{ij}\boldsymbol{x}_j^t + b_i < 0 \\ \text{Bern}(0.5) & \text{if} \sum_j W_{ij}\boldsymbol{x}_j^t + b_i = 0. \end{cases} \tag{1}$$

Here $W$ is the weight matrix, $\boldsymbol{b}$ is a vector of biases, and Bern(0.5) is 0 or 1 with equal probability.

The neuron sums its inputs and turns *on* (*off*) if the sum exceeds (falls short of) a threshold. The network dynamics lead it to some stable fixed point, determined by the connection weights and initial state. Stable fixed points are minima of a generalized energy function:

$$E(\boldsymbol{x}|\boldsymbol{b}, W) = -\frac{1}{2}\sum_{i \neq j} W_{ij}\boldsymbol{x}_i\boldsymbol{x}_j - \sum_i \boldsymbol{b}_i\boldsymbol{x}_i \tag{2}$$

We define a *high pattern number (HPN)* network as one with exponentially many stable states (i.e., $C \sim 2^{\alpha N}$, for some constant $0 < \alpha \leq 1$ with $N$ neurons). A HPN network retains a non-vanishing *information rate* (ratio of log number of coding states to total possible states; $\log(C)/\log(2^N) = \alpha$) even as the network grows in size. Note that, as with the codewords of a good ECC, the stable states of a HPN network cannot be arbitrarily chosen (and theoretical results show that there can be at most O(N) arbitrarily chosen stable states in a Hopfield network[8–10]).

We define a *robust* system as one that can recover the original (or nearest Hamming distance) codeword from a perturbed version with a constant error rate ($p$) in each node. Robust networks must thus tolerate a number of errors ($pN$) proportional to network size, which requires the memory states to be surrounded by attracting basins that grow sufficiently fast with network size.

To our knowledge, no existing neural network model has been shown to combine the two capabilities of exponential capacity and high robustness, in addition to performing self-decoding (see Discussion for more details on previous work).

In the main paper, our focus is on the mapping between ECCs and Hopfield networks, and on the intuition behind why the network combines exponentially-many fixed points with large basins of attraction. Proofs and more technical results are in the SI. In particular, S3–4 prove that general linear codes can be partially mapped onto Hopfield networks but the complete mapping fails, S7–9 prove that expander codes can be mapped to Hopfield networks and provide further details on the construction, S10–11 consider extensions to weaker constraints and noisy updates, and S12 describes a self-organization rule that generates the network.

## 2.1 Using ECCs to construct Hopfield networks with exponentially many well-separated minima and failure of decoding

First, we consider how to directly embed the codewords of a linear ECC into fixed points of a neural network to generate well-separated stable states [11, 12].

Linear ECCs consist of binary codewords of length $K + K' = N$ that satisfy $K'$ constraint parity equations, for a total of $2^K$ codewords separated from each other by up to $K'$ bits. These codes can permit correction of errors on up to $(K' - 1)/2$ bits (see SI S2 for a brief pedagogical overview of ECCs), by an appropriate decoder.

We demonstrate the embedding using the classic (7,4) Hamming code[13], which consists of $2^4$ codewords of length 7, including 3 parity check bits, Fig. 1a (see SI S3 for the embedding of more general linear ECC codewords into neural network fixed points). The 3-bit separation of codewords means that all single bit flips should be corrected by an optimal decoder. The correct codeword is the one that is closest in Hamming distance to the single-bit corrupted state.

The Hamming code can be mapped into the fixed points of an ACA network using a Hopfield network of 7 neurons and 4th-order weights, Fig. 1b [11]: The binary state of one neuron represents one bit (letter) in the codeword while each weight represents a 4-way constraint on the nodes, Figure 1b. By construction, the minimum energy states are the Hamming codewords (SI S3 for a proof).

We achieve the same result more realistically, using only pairwise rather than $K$-th order connections, by introducing a bipartite architecture: Each hidden node can connect pairwise with multiple ($K$) input neurons to enforce the appropriate $K$-th order relationships among the input neurons, Fig. 1c.

Though the well-separated codewords are well-separated minima of the network dynamics, running the dynamics does not appropriately correct a flipped bit, Fig. 1d. Starting one bit-flip from a codeword (at $x_1$), there are many paths for the network to move downhill in energy (energy function in Fig. 1b). One is to appropriately correct the error bit. But, in general, there is only one downhill and correct direction, and many more ($\sim N/2$) ways to move downhill in energy that correspond to steps *away* in Hamming distance (e.g. by flipping bits $x_3$ or $x_7$), producing wrong decoding. For network embeddings of longer Hamming codes, error probability approaches unity (SI S4). Indeed, we expect similar decoding failures for a large class of error-correcting codes (SI S4).

The network's failure should not be surprising, since decoding strong ECCs is computationally hard [14], involving more complex message-passing dynamics than the local sums and pointwise nonlinearities of neural networks. The failure is one of *credit assignment*: the network cannot identify the actual flipped bit, and flips a different bit to move downhill in energy. A successful solution must solve the credit-assignment problem.

## 2.2 Exponential-capacity robust error-correcting ACAs

The central contribution of the present work is to show, we believe for the first time, that it is possible to implement strong error correcting codes with self-decoding in Hopfield networks. Our solution is based on recent developments in graph theory and coding theory that led to ECCs that can be decoded by simple greedy algorithms[7].

These ECCs rely on a constraint structure determined by an *expander graph*. In an expander graph, all small subsets of vertices are connected to relatively large numbers of vertices in their complements. For instance, a subset of 4 vertices, each with out-degree 3, shows good expansion if it projects to 12 other vertices, Figure 2a (top).

We construct a neural network embedding by constructing a bipartite expander graph, consisting of $N, N_C$ input and hidden (constraint) nodes ($N_C \lesseqgtr N$), respectively. Each input node is a neuron while each hidden node is a small network of neurons interacting competitively through inhibition, whose size $K$ does not grow with $N$. Different neurons in a constraint node are connected to the same input subset, but differ in their weights; the competitive interactions serve to separate stable states within the constraint node.

At the level of nodes, connections run between input and constraint layers but not between nodes in a layer, as in a restricted Boltzmann machine [15]. The input and constraint nodes have degrees $z, z_C$ respectively (thus $Nz = N_C z_C$; for expositional simplicity we assume fixed degree; see SI S6-9 for generalization to variable degree). We are interested in *sparse* networks, where the number of

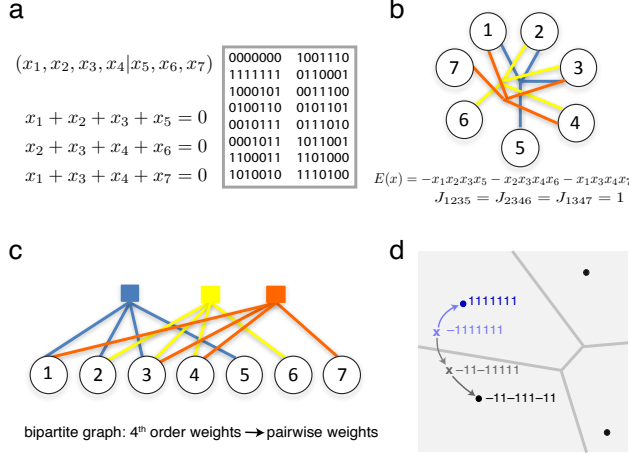

Figure 1: **Successful embedding but incorrect decoding of ECC codewords in Hopfield networks.** (a) Codewords of the (7,4) Hamming code are binary strings of length 7 (right) that satisfy 3 constraint equations (bottom left; sums are modulo 2). (b) The codewords are embedded as the stable states of a Hopfield network with 7 nodes (circles) and 4th-order edges (lines of a single color represent a single 4th-order edge; the edge weight between nodes $i, j, k, l$ is $J_{ijkl}$; binary Hamming states are mapped to $\{-1, 1\}$ in the Hopfield network for notational convenience). Each edge implements one constraint equation. Bottom: The energy function $E$; it is minimized at each codeword. (c) The 4th-order edges can be replaced by conventional pairwise edges if the recurrent network is transformed into a network with a hidden layer. A constraint node (square) is not a single neuron but a small network that implements a parity operation on its inputs (e.g. as in Fig. 3). (d) Schematic showing decoding error. The initial state $-1111111$ (lavender cross), one flip from the closest codeword (1111111; blue) can proceed along multiple trajectories that decrease energy, either to the correct codeword (blue) or to a codeword two flips away (black). Solid lines show regions in state space closest to a particular codeword.

connections the input and hidden nodes each make does not increase with network size. This bipartite graph is a $(\gamma, (1 - \epsilon))$ expander if every sufficiently small subset $S$ (of size $|S| < \gamma N$, for some fixed $\gamma < 1$) of input vertices has at least $(1 - \epsilon)z|S|$ neighbors among the hidden vertices[7, 16–18] ($0 \leq \epsilon < 1$ is some constant, with $\epsilon \to 0$ corresponding to increasing expansion).

Surprisingly, and helpfully in the neural context, sparse random bipartite graphs even with small degree yield good expansion (with $\epsilon < 1/4$ generically) [7, 16–18]. Thus we choose connections between layers randomly, and verify that they are indeed good expanders (SI S5 and Figure S1).

The Lyapunov (generalized energy) function of the network is:

$$E(\boldsymbol{x}, \boldsymbol{h}) = -\left(\boldsymbol{x}^T U \boldsymbol{h} + \boldsymbol{b}^T \boldsymbol{h} + \frac{1}{2}\boldsymbol{h}^T W \boldsymbol{h}\right). \tag{3}$$

Here $\boldsymbol{x}$ and $\boldsymbol{h}$ are vectors of input and constraint neuron activity respectively, $U$ is a sparse matrix with non-zero entries determined by an expander graph, $W$ is a block-diagonal matrix of inhibitory connections and $\boldsymbol{b}$ is a vector of fixed background inputs to the constraint neurons (for further details see SI S8).

The network capacity is exponential, Figure 2b: the number of stable states grows exponentially in the total size of the network (total number of neurons in input and constraint nodes). Moreover, the network dynamics appropriately moves perturbed states back to their closest (in Hamming distance) stable states, correcting a number of errors that increases in proportion to the number of input nodes (and thus also total network size), Figure 2c-e (proofs in SI, S6-9).

Specifically, if each constraint node contains at most $K_{\max}$ neurons (independent of $N$; thus the total network size is $N_{total} \leq N + K_{\max} N_C$), we prove that the expected total number of energy minima and the total number of correctable errors are given by

$$N_{\text{states}} \geq 2^{\alpha N_{total}} \quad \text{and} \quad N_{\text{errors}} \geq \beta N \tag{4}$$

where $\alpha = \frac{(1-\langle r \rangle \hat{z})}{(1+K_{\max}\hat{z})}$ and $\beta = \gamma(1-2\epsilon)$ are both finite constants. $\hat{z} = \frac{\langle z \rangle}{\langle z_C \rangle}$ is the ratio of the average input and hidden layer degrees across nodes ($= z/z_C$ for a regular network) and $-\langle r \rangle$ is the average across nodes of the log ratio of permitted states to all states for a constraint node (SI S8 for more details). The number of minimum energy states is exponential in total network size because $\alpha$ is independent of $N, N_{total}$. We need only be concerned with errors in the inputs because the initial conditions in the constraint nodes are set by relaxation after briefly clamping the input nodes (equivalently, all errors in the constraint neurons are irrelevant/correctable).

Given any per-neuron error probability smaller than $\beta$, the network corrects all errors (with probability $\rightarrow 1$ as $N \rightarrow \infty$). Typical of strong ECCs, the probability of correct inference is step-like: all errors smaller than a threshold size are corrected, while those exceeding the threshold result in failure.

The results of Figure 2 involve an expansion coefficient $\epsilon > 1/4$; constraint nodes that only permit input configurations that differ in the states of at least two neurons (described below); and noise-free (Hopfield-like) update dynamics (beyond the randomness present from the asynchronous nature of the updates). In SI S10 and 11 we extend these results to less stringent conditions on the constraint nodes and show that analogous results, up to small fluctuations around the noiseless stable state, hold for stochastic dynamics in a Boltzmann Machine (Fig. S2).

## 2.3 Compositional structure and dynamics of the robust high-capacity ACA

We next examine the structure and dynamics of this network, to explain how it works. In the section after, we examine why, by virtue of its sparse expander structure, it does not fall prey to the credit-assignment errors exhibited by neural network implementations of Hamming codes.

Within a constraint node, a small network of $K$ neurons with recurrent inhibition attempts to drive the input states to a smaller set of permitted states.[1] Thus, although the network has a restricted Boltzmann machine architecture at the level of input neurons and constraint nodes, the constraint neurons in a given node interact through lateral inhibition. All neurons in one constraint node connect to the same subset of $z_C$ input neurons, Figure 3a, but each will prefer a different configuration of states on these input neurons, as determined by its weights. For our construction, it is important that these preferred configurations differ from one another in at least two bits or two input neuron states. We will show below a simple self-organized way for these well-separated configurations to emerge, but for the moment will simply assume that these hidden node preferred states are at least two bits apart. Then, there are at most $2^{z_C-1}$ preferred states (out of $2^{z_C}$ possible states for the input subset) requiring $K \leq 2^{z_C-1}$ neurons in a constraint node (Figure 3b). Operationally, $z_C$ can be small (between 2 and 6 in Figure 2). For simplicity in Figure 2, 3, we choose the preferred states to be even parity states of the input, by setting weights appropriately (see SI S6, S8-10 for generalizations).

By virtue of their shared inputs, neurons in a constraint node could be viewed as a *glomerulus* (this describes connectivity and need not imply physical clustering as for olfactory or cerebellar glomeruli; it also does not perform amplification as in sensory glomeruli). While input-to-constraint node connectivity is random, connectivity of individual neurons within a constraint node is correlated since these neurons receive the same set of inputs — the network is therefore not fully random.

Taken together, the stable states of the network are compositions or combinations of the preferred input configurations of the different constraint nodes. In the absence of constraints, the input layer has $2^N$ possible states. Each constraint node cuts in half the number of possible states (if preferred patterns in the constraint node are based on parity; see SI S6, S8-10 for generalizations); thus, with $N_C$ constraints there are $2^{N-N_C}$ stable states. Since $N_C$ is a fraction of $N$, the number of stable states is exponential in $N$ (and in total network size; see above). By virtue of the $\geq 2$ separation in patterns per constraint node and the expansion property of the connectivity, the stable states are separated by at least $\gamma N$, which is linear in $N$ (recall that $\gamma$ is the expansion coefficient).

To understand how the network works, first consider the activity of a constraint node. Constraint nodes are conditionally independent of each other given the inputs; thus each constraint node can be studied individually, with its inputs. When a neuron within a constraint node receives an input that exactly matches its preferred configuration, it becomes active and silences the others through strong inhibition, Figure 3a. This is a low-energy state for that node (SI S8), which we will refer to as

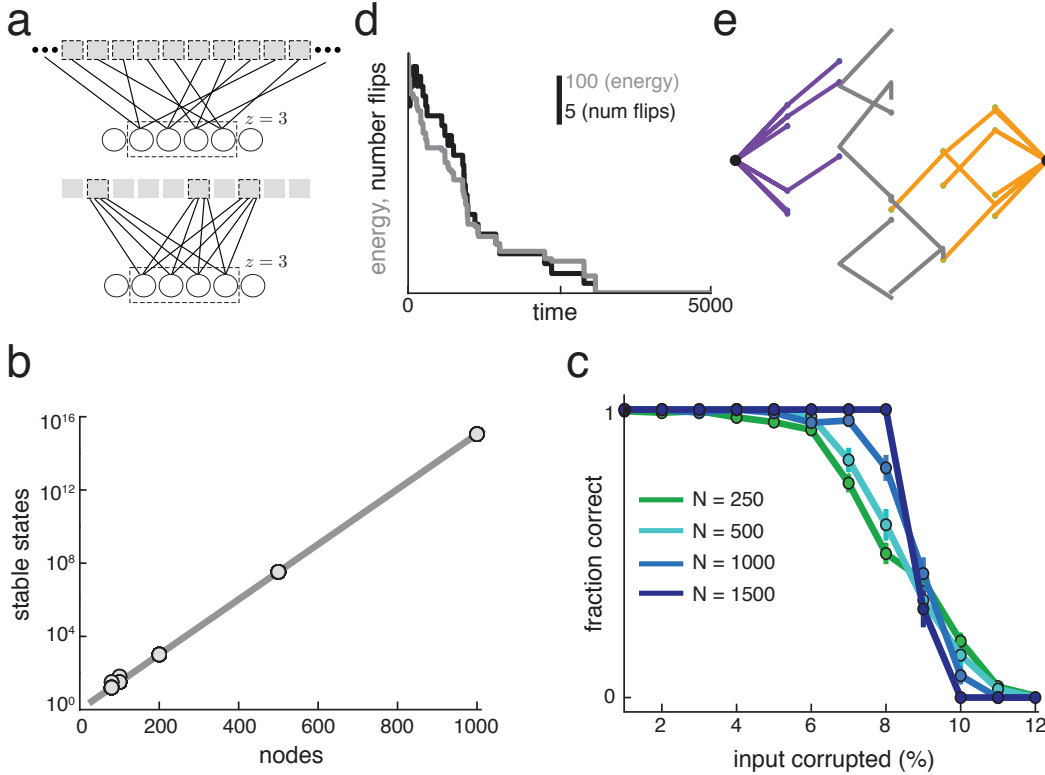

Figure 2: **ACA network with exponential capacity and robust error correction.** (a) Good (top) vs. poor (bottom) expansion in bipartite graphs. Input nodes each send $z = 3$ edges to the constraint layer. Subsets of input neurons (one subset highlighted by dashed line) have many (few) neighbors in good (poor) expanders. (b) Network capacity is exponential. Gray line: Derived theoretical lower bound on number of robust stable states versus the number of input neurons ($N$). Open circles: Number of robust stable states in simulated networks (100 simulations for each point). Since constraint nodes apply parity constraints (see SI S6, S8-10 for generalizations), we numerically calculate the number of stable states as the dimension of the null space of the constraint matrix in the binary field $F_2$. The slight scatter of points reflects occasional duplicate constraints in small random networks (vanishingly rare for large $N$). (c) Fraction of times the network converges to the correct state when a finite fraction of nodes (thus a linearly growing number of nodes with network size) is corrupted. The increasingly sharp transition between recovery and failure (as N increases) is characteristic of ECCs. Error bars: standard error. Minimum of 25 runs for each data point. (d) Energy (gray) and number of node flips (black) over time in an $N = 500$ neuron network with 4% initial corruption, as the network relaxes to the closest stable state. Energy always decreases monotonically, but the number of node flips need not. For panels (b-d), $N_C = 0.95N$, $5 \leq z \leq 10$ and $2 \leq z_C \leq 6$ (see SI S8 for further details). (e) Network state-space trajectories (projected onto 2D space) in a simulated small network. Black dots: two stable states. Different initial states (with 1-5 nodes corrupted, purple, gray, and orange dots) in the vicinity of the stable state to the left and their flows to the stable states. All initial states within a threshold Hamming distance of the original stable state flow to it (purple dots). Those not within that distance (gray and orange dots) flow to other stable states including spurious states. The trajectories that end in the adjacent stable state shown at right are in orange. $N = 18, N_C = 15, z = 5, z_C = 6$.

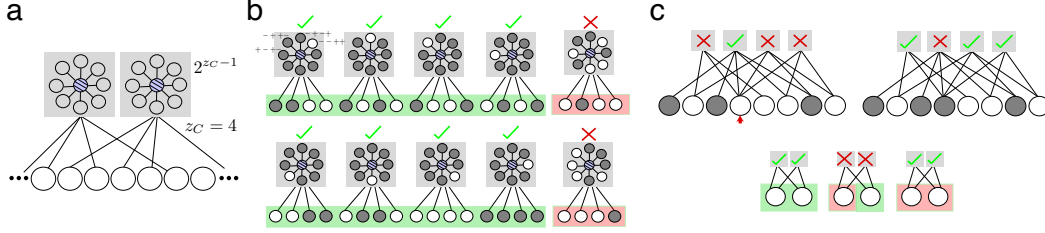

Figure 3: **Architecture and dynamics of a robust exponential capacity ACA.** (a) A constraint node (square box) is a small subnetwork of $2^{z_C-1}$ neurons, with global inhibition within the node (hashed node is inhibitory). Recall that $z_C$ is very small (typically 5-10) and does not change with network size. (b) Single constraint node: All neurons in one constraint node connect to the same inputs, but with different binary weights ($+$'s and $-$'s shown above a few neurons) that determine which input pattern a constraint neuron prefers. E.g. (first example, top row) a constraint neuron with input weights $--++$ prefers 0011 and wins the competition to be active for that input (on state: white fill), silencing the rest (off state: gray). Green check-mark (red cross) shows the constraint node in a (un)satisfied state. Here, even-parity patterns (an even number of 1's) are the only preferred input states. (c) Top left: network with both satisfied and unsatisfied constraints. Flipping an input attached to more unsatisfied than satisfied constraint nodes (red arrow, bottom) lowers the energy of the network when it flips, by flipping the status of all its constraint nodes (top right). Bottom: The problem of poor expansion: if two constraints have overlapping input states, they cannot identify the source of multiple shared errors. A single error in the shared inputs violates both constraints (center), and if a second shared input is corrupted (right) both constraints are satisfied and the iteration process from above fails.

"satisfied", Figure 3b (left, green). If the input exactly matches none of the preferred configurations, more than one constraint neuron will receive equal drive, Figure 3b (right, red); inhibition is strong enough that no more than two among the equally-driven constraint neurons can be active. Such a state corresponds to a higher energy, "unsatisfied" configuration. The overall energy of this network is proportional to the total number of unsatisfied constraint nodes (SI S3 and S9), allowing local energy-minimizing dynamics to successfully correct errors as we describe below.

### 2.4 Credit assignment with expander graph architecture

Heuristically, the network solves credit assignment as follows. For networks with sparse and expansive connectivity, any small set of input neurons shares few common constraint nodes (Fig. 2a and 3c). Thus, if a small fraction of input nodes is wrong, each constraint node will typically receive one or zero wrong inputs. The total number of unsatisfied constraints, which is proportional to the energy of the network, reflects the number of corrupted inputs, aligning the metrics of energy and corrupted input bits. In addition, the specific constellation of unsatisfied constraint nodes for codes on sparse expander networks (but not in standard ECC embeddings into a neural network, including the Hamming code implementations above) is a detailed fingerprint of the pattern of input errors; this fingerprint determines which specific input neurons are in error and should be flipped.

Specifically, for a sufficiently small input error rate ($< \gamma(1 - 2\epsilon)$), a majority of the constraint nodes that receive any corrupted inputs are connected to only one corrupted input. These constraint nodes are appropriately unsatisfied. Next, the network dynamics highly preferentially updates inputs that are connected to more unsatisfied than satisfied constraint nodes (for proofs, see SI, S6-9 and [7]); this process both reduces the energy of the network and the number of corrupted inputs: all unsatisfied constraints connected to the flipped input bit will now be satisfied, and vice versa. The network thus iteratively reduces unsatisfied constraints and, doing so, corrects errors.

### 2.5 Self-organization to robust exponential capacity

Robust exponential capacity requires that preferred patterns at each constraint node differ by at least two bits. The neurons within each constraint node, connected to the same subset of input neurons, can come to prefer sufficiently non-overlapping patterns through a simple self-organization rule. Briefly,

the self-organization rule proceeds by pairing very-sparse random activation of the constraint neurons in a node with random activation of the inputs, together with inhibition within the constraint node. An active constraint neuron wires with strength +1 to co-active inputs and -1 to inactive inputs, in a Hebbian-like one-shot modification. Lateral inhibition prevents constraint neurons from becoming activated by too-similar patterns. We prove that this self-organization rule leads each constraint node to come to prefer a set of input states that differ in at least two entries, and does so in a time that scales only logarithmically in network size (SI S12 and Fig. S3 for further details).

## 3 Discussion

We leveraged recent constructions of low-density parity check codes based on expander graphs (expander codes), which admit decoding by simple greedy algorithms [7, 16], to show that ACA networks with quasi-random connectivity can have and decode exponentially-many robust stable states. In sum, we have constructed simple ACA networks with capacity and robustness comparable to state-of-the-art codes in communications theory, moreover with the decoder built into the dynamics.

Spin glasses (random-weight symmetric Hopfield networks) possess exponentially many (quasi)stable fixed points [19] but these energy minima have not been shown to have large basins of attraction. Instead, most minima are not observable in numerical explorations [20], suggesting small basins of attraction. Hopfield networks designed for constraint satisfaction problems have $\sim 2^{\sqrt{N}}$ stable states [21–23]; a recent construction with hidden nodes has capacity that is exponential in the ratio of the number of hidden to input neurons[24]; and a network based on a sparse bipartite graph with integer-valued neural activity shows exponentially many stable states [25]. However, in all cases the fraction of correctable errors either vanishes with network size or is negligible to begin with. Thus none of these networks are robust to noise.

The first attempts to link ECCs with spin glasses and Hopfield networks were by Sourlas [11, 12], where the goal was to use the relationship to decode noisy messages. However, these studies construct a separate neural network (with different weights and energy function) for decoding inputs in the neighborhood of each codeword. Thus, each network carries out decoding specific to a particular input and does not possess exponentially-many stable states with large basins of attraction that appropriately decode states near any of the robust stable states.

Near-exponential capacity was recently realized in Hopfield networks with clique structures. Hillar & Tran [26] construct a network whose stored patterns correspond to the cliques of an abstract graph, with capacity $C =\sim e^{\alpha \sqrt{N}}$. Fiete et. al [27] divide a network of $N$ neurons into $N/\log(N)$ non-overlapping binary switches, for capacity $C =\sim e^{N/\log(N)}$. These networks have multiple nice features: they have large basins of attraction, converge rapidly and are easy to construct. However, their information rate ($\rho \equiv \ln(C)/N$) still vanishes asymptotically in $N$ and thus does not match good error-correcting codes. They are also susceptible to sub-linear sized adversarial patterns of error. Our network, which does not break into distinct sub-linear sized subnetworks due to the intermeshed expander graph structure is not similarly susceptible.

The network has a two-layer restricted Boltzmann machine-like architecture and can be represented as a factor graph (constraint modules are factors), undirected graphical model (clique potentials are indicator functions on visible neurons in a constraint) and Bayes net (in several different ways). Our network can be made structurally similar to the Bayes nets used to decode ECCs [28, 29] by adding an input layer with fixed noisy inputs and slightly rewriting constraint modules. The primary difference is dynamical rather than structural. Expander codes permit simple local decoding [7] and thus we are able to use Hopfield dynamics as the decoding rule in the network. These dynamics are significantly simpler than belief propagation (BP). In general, codes that can be decoded by BP do not admit simple network decoding by the energy-based Hopfield rule and we do not expect general ECC decoding to be performed by Hopfield dynamics. In the SI we make preliminary attempts to characterize when ECCs can be mapped to neural networks. It will be interesting, in future work, to determine when the mapping is possible and to analyze a broader set of ECCs as neural networks.

Hopfield networks of $N$ neurons cannot store for full recall more than $O(N)$ *arbitrary* patterns [8, 9]. The patterns stored in our exponential capacity network are not arbitrary and satisfy a conjunction of many sparse constraints. Given these restrictions, how might these (or any other supralinear capacity networks[26, 27]) be used?

One possibility is that the networks are used in the traditional Hopfield network sense, as content-addressable memories to recall entire input patterns, but with very high capacity for inputs with appropriate structure. Indeed, natural inputs that are stored well by brains are not random. For example, natural images are generated from latent causes or sources in the world, each imposing constraints on a sparse subset of the retinal data we receive, and might be reasonable candidates to store in such a Hopfield network [30]. Alternatively, processing the data to decorrelate lower moments [31–33] while preserving or adding information in higher moments might produce appropriate structure. It is an open question what stimuli might either be already described within this framework or be naturally transformed to acquire the appropriate structure.

A second possibility is to use these networks as high-capacity pattern labelers or locality-sensitive hash functions. Here input patterns in a very high-dimensional space (possibly neocortex) are mapped to the exponentially-many stable states of a HPN attractor network (possibly hippocampus), which serve as memory labels for the patterns. The connectivity matrix can be constructed in a simple, online, Hebbian way. When presented with a noisy version of an input pattern, the memory network robustly retrieves the correct label (and maintains it in the absence of input). Such a pattern labeler can be used for recognition or familiarity detection, template matching, classification, locality-sensitive hashing and nearest neighbor computations, and could also play a role in memory consolidation and learning conjunctive representations.

Our network is structurally simple, apart from the modular organization of the constraints: connectivity is pairwise and random, and weights have low dynamic range (in fact they are binary). This simplicity suggests that such error-correction strategies may be used in the brain, and it will be interesting to look for signatures of them in neural data. One signature is that the network has many more constraint cells than input cells. Constraint cells are very sparsely active because only those few which receive their preferred input are driven, along with some others that are transiently active to correct errors. The smaller number of constraint cells that receive their preferred inputs, together with the input cells that carry the actual representation will respond stably. Consequently, representations are predicted to contain a dense but small stable core, with many other neurons that are transiently active and not reliably responsive across repeats of a pattern (because errors might be different each time). This is reminiscent of observations in place cell populations [34] and the sparse, heavy-tailed distribution of population activities across hippocampus and neocortex [35, 36]. A second signature of these strategies is that representations in the stably active neurons should have decorrelated second-order (i.e. pairwise) statistics but contain structure in higher-order moments that the network exploits for error correction.

Our results generalize along multiple directions, and provide several insights into the structure of sparse neural networks. We show many other results in the supplement, including that: 1) The same principles carry over to stochastic Boltzmann machines (SI S11). 2) Sparse random higher-order Hopfield networks, even without hidden nodes, are isomorphic to expander codes (SI S7). Thus, higher-order Hopfield networks generically have exponential capacity with high robustness, if they are sparse and random. Higher-order Hopfield networks are used to model disordered systems in physics, often under the name of $p$-spin infinite-range models[37]. The connection to expander codes may lead to further insights into these models. 3) Energy-based decoding is likely to generically fail on dense networks, suggesting that sparsity may be necessary, not merely sufficient, for high-capacity neural networks (SI S4). 4) We extend the finding that the total capacity (the product of information per pattern times number of patterns) of any Hopfield network with pairwise connections is theoretically bounded at $O(N)$ arbitrary dense patterns [8–10] and $O(N^2)$ arbitrary sparse patterns[38–40], to show that these bounds also hold for architectures with hidden nodes (SI S13).

Expander graph neural networks combine many weak constraints and exploit properties that are rare in low dimensions but generic in high dimensions, both common tropes in modern computer science and machine learning[41–44]. Expander graphs have found widespread recent use in designing algorithms[17, 7, 45]. Because neural networks are large and sparse, and because large sparse random networks are good expanders[17, 18], expander architectures may provide broader insight into the computational capabilities of the brain. Our results may bridge problems in neural networks and a growing body of powerful expander-graph-based techniques and algorithms in computer science.

**Code availability**    All simulations were run using scripts written in Python 2.7, along with standard packages. Code is available at https://chaudhurilab.ucdavis.edu/code.

**Acknowledgments**

We are indebted to Peter Latham for extensive and thought-provoking comments and discussion, as well as suggestions for improving the exposition of our results. We are grateful to Yoram Burak, David Schwab, and Ngoc Tran for many helpful discussions on early parts of this work, and to Yoram Burak and Christopher Hillar for comments on the manuscript. IF is an HHMI Faculty Scholar, a CIFAR Senior Fellow, and acknowledges funding from the Simons Foundation and the ONR under a YIP award. Part of this work was performed by RC and IF in residence at the Simons Institute for the Theory of Computing at Berkeley, where RC was supported by a Google Fellowship.

## Footnotes

[1]Within-node inhibition can be replaced by common global inhibition across all neurons and constraint nodes, slowing convergence dynamics but not affecting the overall quality of the computation.

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
