[Supplementary Material · chaudhuri_fiete_supp_info_ecc_exp_hopf_neurips.pdf]

# Supplementary Information

In order to make the SI self-contained, we repeat brief portions of the main text here. S1 and S2 provide short introductions to Hopfield networks and error-correcting codes respectively. S3 demonstrates that the states and energy minima of error-correcting codes can be mapped onto the states of higher-order Hopfield networks. S4 shows that the mapping described in S3 is only partial, in that error-correcting codes cannot, in general, be decoded by neural network dynamics. S5 describes sparse bipartite expander graphs. S6 describes the construction of expander codes by Sipser & Spielman[1]. S7 shows that expander codes are isomorphic to sparse higher-order Hopfield networks. S8 and S9 describe the construction of expander-graph-based pairwise Hopfield networks that have exponentially-many global energy minima and large basins of attraction. S10 and S11 consider the case of weaker constraints and noisy state updates, respectively, and show that exponential capacity and good error-correction still approximately hold. S12 describes a self-organization rule that allows construction of the exponential-capacity network. S13 shows that standard bounds on the capacity of Hopfield networks for arbitrary patterns still apply when the networks have hidden nodes.

## Contents

# S1    Hopfield networks and Boltzmann machines

We consider networks of $N$ binary neurons. At a given time, $t$, each neuron has state $\boldsymbol{x}_i^t=0$ or 1, corresponding to the neuron being inactive or active respectively. The network is defined by an $N$-dimensional vector of biases, $\boldsymbol{b}$, and an $N \times N$ symmetric weight matrix $W$. Here $\boldsymbol{b}_i$ is the bias (or background input) for the $i$th neuron (equivalently, the negative of the activation threshold), and $W_{ij}$ is the interaction strength between neurons $i$ and $j$ (set to 0 when $i = j$).

Neurons update their states asynchronously according to the following rule:

$$\boldsymbol{x}_i^{t+1} = \begin{cases} 1 & \text{if } \sum_j W_{ij}\boldsymbol{x}_j^t + b_i > 0 \\ 0 & \text{if} \sum_j W_{ij}\boldsymbol{x}_j^t + b_i < 0 \\ \text{Bern}(0.5) & \text{if} \sum_j W_{ij}\boldsymbol{x}_j^t + b_i = 0 \end{cases} \tag{1}$$

Here Bern(0.5) represents a random variable that takes values 0 and 1 with equal probability.

Hopfield networks can also be represented by an energy function, defined as

$$E(\boldsymbol{x}|\boldsymbol{\theta}, W) = -\frac{1}{2}\sum_{i \neq j} W_{ij}\boldsymbol{x}_i\boldsymbol{x}_j - \sum_i \boldsymbol{b}_i\boldsymbol{x}_i \tag{2}$$

The dynamical rule is then to change the state of a neuron if doing so decreases the energy (and to change the state with 50% probability if doing so leaves the energy unchanged).

We also consider Boltzmann machines, which are similar to Hopfield networks but have probabilistic update rules.

$$\boldsymbol{x}_i^{t+1} = \text{Bern}(p)$$
$$\text{where } p = \frac{1}{1 + e^{-\beta(\sum_j W_{ij}\boldsymbol{x}_j^t + b_i)}} \tag{3}$$

The probability of a state $\boldsymbol{x}$ in a Boltzmann machine is $p(\boldsymbol{x}) \propto e^{-\beta E(\boldsymbol{x})}$, where $E(\boldsymbol{x})$ is defined as in Eq. 2 and $\beta$ is a scaling constant (often called inverse temperature). Note that the Hopfield network is the $\beta \to \infty$ limit of a Boltzmann machine.

Hopfield networks with $N$ neurons and $\sim N^2$ pairwise analog (infinite-precision) weights trained with simple learning rules can learn and exactly correct up to $N/(2\log(N))$ random binary inputs[2] or imperfectly recall $0.14N$ states (with residual errors in a small fraction of neurons)[3]. With sparse inputs or better learning rules, it is possible to store and robustly correct $\sim N$ states[4–7]. Independent of learning rule and architecture, the capacity of Hopfield networks with pairwise connections is theoretically bounded at $\sim N$ arbitrary states[8–10]. Thus, achieving high pattern capacity requires that the memory states have special structure.

# S2    Error correcting codes

Given a string of variables (message) to be transmitted, an error correcting code (ECC) adds redundancy to allow the message to be recovered despite added noise. A parity check

code over some set of variables $x_i$, where each $x_i \in \{0,1\}$, is defined by a set of constraints, $\sum x_i = 0$, where the sums are taken modulo 2. Thus each constraint restricts its participating variables to some set of acceptable states.

A classical example of such an ECC is the (7,4) Hamming code[11], which is defined by considering 4-bit messages and adding 3 parity-check bits to the message, defined as

$$x_5 = x_1 + x_2 + x_3$$
$$x_6 = x_2 + x_3 + x_4$$
$$x_7 = x_1 + x_3 + x_4 \tag{4}$$

For example, instead of transmitting the message 0101, three additional check bits are added on and the message transmitted is 0101101. Thus there are $2^4$ possible correct messages (see Fig. 1); these possible message are called codewords. If a message is received that does not correspond to a codeword, it is mapped to the closest codeword, thus correcting errors. The parity check bits are chosen so that any two codewords are separated by the state of at least three bits. Thus, if a transmitted message is received where a single bit is flipped, the Hamming code can recover the original message.

The *distance* of a code is the separation between codewords, and is twice the number of errors that can be corrected. The *rate* of a code is the number of information bits transmitted per message bit. The (7, 4) Hamming code has a distance of 3 and a rate of 4/7.

Rather than seeing a codeword as the combination of a desired message and a set of added check bits, the parity check bit equation can be reframed as a set of 3 constraints on 7 variables:

$$x_1 + x_2 + x_3 + x_5 = 0$$
$$x_2 + x_3 + x_4 + x_6 = 0$$
$$x_1 + x_3 + x_4 + x_7 = 0 \tag{5}$$

The codewords are the states that satisfy these equations. Thus the Hamming codewords occupy a 4 dimensional subspace of a 7 dimensional space.

The constraint structure of a code can be represented as a bipartite graph[a], with one set of variable nodes and another set of constraint nodes. This is shown for the (7,4) Hamming code in Fig. 1, with 7 variables and 3 constraint nodes (each corresponding to an equation). Analyzing and constructing codes from a graph-theoretic perspective has been a very fruitful area of research for the last three decades[12].

Considering longer blocks of bits allows the construction of codes with better performance, meaning that they either have a larger distance between codewords (i.e., correct more errors) or send information at a higher rate (i.e., more efficiently), or some combination of the above.

The (7,4) Hamming code is one example of a family of Hamming codes of increasing length. For each value $k \geq 2$, there exists a Hamming code of length $N = 2^k - 1$ with $k$ constraints. The code conveys $2^k - k - 1$ bits of information and can correct one error[11,13]. To construct the constraints, first express each variable / message bit in binary (ranging from 1 to $2^k - 1$). Then the $j$th constraint is the sum of all variables that have the $j$-th bit

---

[a]A bipartite graph is a network with two sets of nodes. Nodes in each set connect only with nodes from the other set.

set in their binary expansion. For example, the first constraint sums up bits 1, 3, 5, 7 and so on, and the second constraints sums up bits 2, 3, 6, 7, 10, 11 and so on[b].

# S3 Mapping between general linear codes and higher-order Hopfield networks

**Claim 1.** *A parity check code can be mapped onto Hopfield networks whose neurons, $s_i$, take states in $\{-1, +1\}$ by mapping binary state 0 to +1 and binary state 1 to -1.*

*Proof.* The parity check constraints $\sum_{i \in C_i} x_i = 0$ can be reexpressed as products $\prod_{i \in C_i} s_i = 1$. These can be used to define an energy function

$$E(\boldsymbol{s}) = - \sum_{C_i} \prod_{i \in C_i} s_i \tag{6}$$

This energy function takes its minimum energy values if and only if all the constraints are satisfied and the energy increases as the number of the violated constraints increases:

$$E(\boldsymbol{s}) = E_{min} + 2N_{VC}(\boldsymbol{s}),$$

where $N_{VC}(\boldsymbol{s})$ is the number of violated constraints in state $\boldsymbol{s}$. Thus the minimum energy states of this network are the codewords of the corresponding parity-check code. $\square$

Note that a mapping also exists to Hopfield networks where neurons take states in $\{0,1\}$, but the energy function is slightly more complex.

Continuing our (7,4) Hamming code example, a Hopfield network with the same minimum energy states as the codewords of the (7,4) Hamming code has energy function:

$$E = -s_1 s_2 s_3 s_5 - s_2 s_3 s_4 s_6 - s_1 s_3 s_4 s_7. \tag{7}$$

This mapping involves higher-order edges, meaning edges that connect more than 2 neurons. It has been known for a while that the codewords of ECCs can be mapped to higher-order Hopfield networks[14;15]. However, we make the further observation that by adding hidden neurons (as we do for our exponential capacity Hopfield network construction later), an equivalent network can be constructed with pairwise interactions.

# S4 Energy-based decoding for Hamming and other codes

For energy-based decoding, we require that the number of violated constraints can serve as a local error signal, meaning that we can decide to flip a neuron based purely on whether doing so reduces the number of violated constraints. In this section we first prove that such decoding generically fails for Hamming codes, and then provide a heuristic argument for why such decoding requires codes where each variable participates in only a small number of constraints (i.e., the bipartite graph representation is sparse), and where small sets of variables do not share many constraints in common.

---

[b]Note that when applied to $k = 3$ this yields the $(7, 4)$ Hamming code from this section and from Figure 1 up to a relabeling of variables (there are slightly different conventions for the order in which the variables are labeled).

## Hamming code decoding

**Claim 2.** *Energy-based (greedy) decoding of a length $N$ Hamming code produces equal or lower energy when changing the state of any one of $N/2 - 1$ variables, but leads away from the nearest codeword in these cases; it both produces equal or lower energy and leads toward the nearest codeword only when changing the state of one specific variable.*

*Proof.* Consider the family of Hamming codes described in Section S2. The construction implies that any given variable has a fixed probability $p$ of participating in each constraint, and $p \approx 1/2$ ($p$ is not exactly half because there is no variable that participates in 0 constraints, but the difference between $p$ and $1/2$ shrinks as N gets larger).

Now consider starting at a codeword and corrupting the state of a randomly-chosen variable, which we call $x_i$ (note that all states in a Hamming code are either a codeword or at a Hamming distance of 1 from a codeword). This makes some set $R$ of constraints unsatisfied, where $|R| \sim \text{Bern}(N, p)$ and is $N/2$ on average. To perform energy-based decoding, we consider the effect of flipping a neuron, $x_j$, on the number of unsatisfied constraints. We already know that flipping $x_i$ back to its original state will make all constraints satisfied. Flipping any of the remaining neurons ($j \neq i$) will lead the system away from the nearest codeword. Therefore, if the system is to perform accurate decoding, a different neuron ($j \neq i$) should not flip, which requires that flipping it ought to result in a higher-energy state. We see next that this does not hold.

If $x_j$ is connected to a set $S$ of constraints, then flipping $x_j$ will change the state of constraints in $S \cap R$ from unsatisfied to satisfied and change those in $S \setminus R$ from satisfied to unsatisfied. Thus the energy of the state with $x_j$ flipped is determined by $|S \setminus R| - |S \cap R|$. Since these sets are chosen independently and with $p = 1/2$, on average $|S \setminus R| = |S \cap R|$. At least 50% of possible flips have $|S \cap R| \geq |S \setminus R|$ and these lead to states that have equal or lower energy. Thus about $N/2$ possible directions lead to states that have equal or lower energy (but are further away from the original codeword), and only 1 of these leads to the desired codeword. $\qquad\square$

Note that here the problem is the overlap between the constraints connected to the variables $x_i$ and $x_j$, suggesting we would like this to be small. Also note that, in this case, gradient descent (i.e., picking the neighboring state with *lowest* energy, not just any state with lower energy) would lead in the right direction, but this is a consequence of our initial state being next to a codeword / global minimum and is not generic; moreover, finding the steepest direction is a harder computational problem that is not solved by Hopfield network update dynamics.

## Local energy-based decoding of other codes

We next heuristically argue that for good local energy-based decoding, a code must be sparse (i.e., each variable participates in a sparse subset of constraints) and that variables should not share too many constraints in common.

First consider a code of length $N$ in which $O(N)$ errors can be corrected by an appropriate decoder[c], and where each variable participates in a fraction $pN$ of constraints. Start at a

---

[c]Note that in the Hamming code, only 1 error can be corrected.

codeword and corrupt (flip) a set of $\alpha N$ bits, which we call $E$. In the absence of special structure, $E$ contacts a number of constraints that grows as $N^2$, and thus even for very small $\alpha$ and $p$, all possible constraints will be connected and will receive multiple edges from the nodes in $E$. Thus, their states will be approximately random.

Now flipping a node that is outside of $E$ will change the state of a set of $pN$ constraints, which we call $T$. Satisfied constraints in $T$ will change to unsatisfied and vice versa. The new state will have lower energy if $T$ contained more unsatisfied than satisfied constraints, which will happen with about 50% probability. Thus, there are many variables outside of $E$ that would lead to lower energy states when flipped, and energy-based decoding will in general not recover the nearest codeword.

Next, consider codes of length $N$ where each variable participates in a small number of constraints that does not grow with $N$. As before, consider a set of error nodes $E$, which will be connected to some small set of constraints, $S$. Some subset of constraints in $S$ are unsatisfied. For accurate energy-based decoding to be possible, flipping variables outside of $E$ should increase the number of unsatisfied constraints while flipping variables in $E$ should decrease the number of unsatisfied constraints. For both cases, this is determined by the overlap of connected constraints between the variable that will be flipped and the nodes in $E$. First, consider flipping a node $x_i$ outside $E$. $x_i$ is connected to some set of constraints $T$. All constraints in $T \setminus S$ will switch from satisfied to unsatisfied, and some subset of constraints in $T \cap S$ will switch from unsatisfied to satisfied. Thus, we would like $T \cap S$ to be as small as possible (and $T \setminus S$ to be large). Similarly, consider flipping a node $x_i$ inside $E$, which is connected to some set of constraints $T \subset S$. Constraints in $T$ that are only connected to $x_i$ and not to other members of $E$ will become satisfied, while a subset of constraints in $T$ that receive multiple edges from $E$ will become unsatisfied. As before, we wish this subset to be small and the number of constraints in $T$ that are only connected to $x_i$ to be large. Thus we wish the overlap of constraints between $x_i$ and $E \setminus x_i$ to be small.

This argument suggests that codes decodable by a local energy-based rule should be sparse and that small sets of variables should not share many constraints in common. In the next section we describe expander graphs, which were used by by Sipser & Spielman[1] to formalize this notion and construct easily-decodable codes.

In summary, for the general case of linear ECCs embedded in a Hopfield network, while the constructed networks have the right energy minima (codewords), the dynamics do not perform optimal decoding (i.e. error-correction). Optimal decoding should map each corrupted codeword to the most likely original codeword, which for IID noise at each variable, corresponds to the nearest codeword (in Hamming distance, meaning the codeword reached by the fewest variable flips from the current state). By contrast, the Hopfield network energy-based dynamics flips a neuron if doing so reduces the number of violated constraints (which is proportional to the energy). While the nearest codeword has lower energy than the current state, energy-based decoding will not necessarily guide the state to the right codeword. The network may end up in a codeword that is not the nearest codeword (as happens for the Hamming code network implementation) or it may get stuck in a local energy minimum (for codes other than the Hamming code). Thus while it is easy to write down an energy function that returns the codewords as minimum energy states, most error-correcting codes cannot be decoded by a local dynamical rule.

# S5    Expander graphs

Expansion is a property of a graph (i.e., network) where small sets of nodes have a large number of neighbors (i.e., connected nodes). In the context of coding theory, sparse expander graphs allow variables to not share many constraints in common, allowing for local energy-based decoding.

There are various ways to formalize the notion of expansion. We consider bipartite graphs, meaning graphs that are divided into two sets of nodes, with connections between these sets but no connections within a set.

**Definition 1.** *Consider an undirected bipartite graph with $N$ nodes in an input layer and $N_C \sim N$ nodes in a hidden layer, which we call a* constraint *layer. Assume that input nodes have connections drawn from some degree distribution with degree $z$ such that $z^{min} \leq z \leq z^{max}$. Similarly, constraint nodes are drawn from a distribution with $z_C < z_C^{max}$. Such a graph is a $(\gamma, (1-\epsilon))$ expander if every set of nodes $S$ in the input layer with $|S| \leq \gamma N$ has at least $(1-\epsilon)|\delta(S)|$ neighbors, where $\delta(S)$ is the set of edges connected to nodes in $S$, and $|\delta(S)|$ is the number of edges in this set.*

Thus, in an expander graph, small subsets of variables ("small" is determined by $\gamma$) participate in proportionately large sets of constraints ("large" is determined by $1-\epsilon$). Note that if the edges emerging from $S$ each target disjoint nodes, then $\epsilon = 0$. Thus, $\epsilon \to 0$ corresponds to increasing expansion.

Expander graphs can be constructed in various ways, but sparse random bipartite graphs are generically expander graphs[1;16]. We use the following lemma from Luby et al. (2001).

**Lemma 1.** *Let $B$ be a bipartite graph, with nodes divided into $N$ left nodes and $N_C$ right nodes. Suppose that a degree is assigned to each node so that all left nodes have degree at least five, and all right nodes have degree at most $C$ for some constant $C$. Suppose that a random permutation is chosen and used to match each edge out of a left node with an edge into a right node. Then, with probability $1 - O(1/N)$, for some fixed $\gamma > 0$ and $\epsilon < 1/4$, $B$ is a $(\gamma, (1-\epsilon))$ expander.*

In our simulations we generate all graphs randomly by picking edges to connect pairs of variable and constraint nodes subject to the constraints on the degree distributions (see Figure S1 for numerical estimates of expansion in these graphs).

# S6    Irregular expander codes with general constraints

The following analysis is based on Sipser & Spielman[1] and Luby et al.[16], with slight generalization to consider the case of general rather than parity constraints (note that in this case the capacity results hold in expectation; see SI 8.2). We consider an undirected bipartite graph with $N$ nodes in the input layer and $N_C < N$ nodes in the hidden layer, which we call a *constraint* layer. Assume that input nodes have connections drawn from some degree distribution with degree $z$ such that $z^{min} \leq z \leq z^{max}$. Similarly, constraint nodes are drawn from a distribution with $z_C < z_C^{max}$. We assume that such a graph is an expander for some

Figure S1: **Estimates of network expansion.** Each plot shows the ratio of the number of neighbors to the number of edges for sample subsets of neurons drawn from the networks used to generate Figure 2. This ratio corresponds to $(1 - \epsilon)$ in the main text. Dashed line shows $(1 - \epsilon) = 0.75$, which is the theoretical lower bound on expansion required for good error correction.

$\gamma > 0$ and with $\epsilon < 1/4$. Note that the lemma in the previous section guarantees that such graphs can be constructed randomly.

We consider a code defined on this bipartite graph, where input nodes can take values in $\{0, 1\}$ and constraint nodes are satisfied by some configurations of their inputs, with these preferred configurations differing on the state of at least 2 nodes. In the simplest case, these are parity constraints and thus the number of codewords grows as $2^{N-N_C}$ (for discussion of capacity with more general constraints see SI 8.2).

The bit-flip algorithm of Sipser & Spielman[1] proceeds by flipping the state of any variable if doing so reduces the number of unsatisfied constraints. As we review below, the algorithm can correct a number of errors proportional to $N$ around a state where all constraints are satisfied.

Given two configurations of the input nodes, $Q_1$ and $Q_2$, define the distance $d(Q_1, Q_2)$ to be the number of nodes at which they differ (this is just the Hamming distance). Let the set $\Delta(Q_1, Q_2)$ contain the list of the nodes at which $Q_1$ and $Q_2$ differ.

To prove that the network performs good decoding / pattern completion, we consider a network where the input nodes are in state $Q$, where $Q$ differs from some state $Q_{sat}$ where all constraints are satisfied. Let $E = \Delta(Q_{sat}, Q)$ be the set of nodes at which they differ. We will show that if $d(Q_{sat}, Q) < \frac{z^{min}}{z^{max}}(1 - 2\epsilon)\gamma N$, then the network dynamics converges to $Q_{sat}$ (for constant degree, as reported in the main text, this is $(1 - 2\epsilon)\gamma N$; moreover, since $\epsilon < 1/4$, the number of correctable errors is at least $\gamma N/2$).

By assumption $d(Q_{sat}, Q) = |E| \leq \gamma N$. Define $U(E)$ to be the unsatisfied constraints connected to $E$ and $S(E)$ to be the satisfied constraints connected to $E$. The neighbors of $E$ are $N(E) = U(E) \cup S(E)$ (and the number of neighbors is $|N(E)|$). Define a *unique neighbor* of $E$ to be a constraint node that is only connected to one node in $E$, and Unique($E$) to be the set of such neighbors. We start by lower bounding $|$Unique$(E)|$ using a counting argument.

Since $|E| \leq \gamma N$, the expansion property holds, and $E$ has at least $(1-\epsilon)|\delta(E)|$ neighbors (recall that $\delta(E)$ is the number of edges leaving the set $E$). Thus, at least $(1-\epsilon)|\delta(E)|$ of the edges in $\delta(E)$ go to different constraint nodes. There are $\epsilon|\delta(E)|$ remaining edges, meaning that at most $\epsilon|\delta(E)|$ constraint nodes can receive more than one edge and the remainder receive exactly one edge from $E$ and are unique neighbors. Consequently

$$|\text{Unique}(E)| \geq (1-\epsilon)|\delta(E)| - \epsilon|\delta(E)| = (1-2\epsilon)|\delta(E)|. \tag{8}$$

The number of unique neighbors of $E$ is determined purely by graph connectivity and not by the particular constraints imposed. The expansion property guarantees that the number of unique neighbors is large.

Next, we translate the $|$Unique$(E)|$ into a bound on $U(E)$, the set of unsatisfied constraints. In the simplest case, the acceptable states for each constraint differ on at least 2 variable nodes (we weaken this assumption in the following section). Any constraint $C \in \text{Unique}(E)$ is connected to only one corrupted variable node and is thus violated (but is one flipped bit away from being satisfied). Consequently,

$$|U(E)| \geq (1-2\epsilon)|\delta(E)|. \tag{9}$$

Note that since $\epsilon > 1/2$, $|U(E)| > 0$ and thus some constraints are unsatisfied. This result guarantees that any states which satisfy all constraints must differ on $> \gamma N$ nodes. This

follows because the argument above applies to any state $Q$ such that $d(Q, Q_{sat}) \leq \gamma N$ nodes, where $Q_{sat}$ is a satisfied state. Consequently $d(Q, Q_{sat}) \leq \gamma N$ implies that $Q$ does not satisfy all constraints.

The randomized construction allows us to construct networks with $\epsilon > 1/4$, guaranteeing that $|U(E)| > |\delta(E)|/2$. Thus at least half the edges leaving $E$ target unsatisfied constraints, meaning that at least one node in $E$ is adjacent to more unsatisfied than satisfied constraints. Hence there is always a node whose state is energetically unfavorable and that will eventually switch.

This statement is stronger than the claim that states with satisfied constraints are separated by $\gamma N$. While states with all satisfied constraints are minima, it might have been the case that there also exist local minima with unsatisfied constraints within this $\gamma N$ radius. For example, if all the error nodes were connected to more satisfied than unsatisfied constraints, then flipping any single node would increase the energy function. The corresponding input state would be a local minimum and would be an acceptable steady-state for the neural network (a non-local decoding algorithm could circumvent this by searching over a wider set of neighbors). Note that local minima might still exist, but not within a radius of $\gamma N$ of a state with all satisfied constraints.

Thus far we have shown that minima of the energy function must differ on the states of $> \gamma N$ variables, and that there is always at least one error node that it is favorable to flip if the number of errors is less than $\gamma N$. However we have not shown that the network dynamics converges to the right energy minimum (it is also possible to flip non-error nodes). To establish correct convergence, consider a network state on decoding step $t$, $Q(t)$, such that $\Delta(Q(t), Q_{sat}) = E(t)$. Let $U(t)$ be the set of unsatisfied constraints at time $t$, and note that the network dynamics always decreases the number of unsatisfied constraints, so $U(t+1) < U(t) < \cdots < U(0)$.

We require that $|E(0)| < \frac{z^{min}}{z^{max}}(1-2\epsilon)\gamma N$ (recall that $z^{min}$ and $z^{max}$ bound the degree of nodes in the input layer; this ratio is 1 for the constant degree nodes we discuss in the main text). Each variable in $E$ sends out a maximum of $z^{max}$ edges, and a constraint node can only be unsatisfied if it receives one of these edges. Thus

$$|U(t)| \leq |U(0)| \leq |\delta(E(0))| \leq z^{max}|E(0)| < z^{min}(1-2\epsilon)\gamma N \tag{10}$$

On the other hand, energy minima are separated by a distance of at least $\gamma N$. Thus if the network starts in a state with distance $|E| < \gamma N$ from $Q_{sat}$ and ends up at another energy minimum, it must pass through a state with $|E| = \gamma N$. By Eq. 9, this intermediate state has at least $(1-2\epsilon)|\delta(E)| \geq z^{min}(1-2\epsilon)\gamma N$ violated constraints, which contradicts Eq. 10. In summary, while the network dynamics may (transiently) increase the size of $E$, it does not increase the number of violated constraints. Consequently, if we start in a state with fewer than $\frac{z^{min}}{z^{max}}(1-2\epsilon)\gamma N$ errors we will always remain in a state with fewer than $\gamma N$ errors. In this case Eq. 9 guarantees that there is always a node which will change its state, and doing so reduces the number of violated constraints by at least one. The network will thus converge in a time bounded by the product of the number of violated constraints and the time it takes each variable node to flip.

Relatedly, note that the Hamming codes we considered previously have expansion that goes to 0. For example, in a Hamming code with $N$ variables and $k$ constraints (see S2 for

the definition of the family of $(N, k)$ Hamming codes), consider the set $\hat{S}$ of variables that participate in $k-1$ constraints. There are $k$ such variables, and each sends out $k-1$ edges. The set as a whole sends out $|\delta(\hat{S})| = (k-1)k$ edges and is connected to every constraint node so has $|N(\hat{S})| = k$ neighbors. Consequently, $|N(\hat{S})|/|\delta(\hat{S})| \to 0$ as $k \to \infty$. For expansion, we require that for some $\gamma > 0$, any set $S$ of size $|S| < \gamma N$ has at least $(1-\epsilon)|\delta(S)|$ neighbors. The set $\hat{S}$ has size $O(\log(N))$ and thus, no matter how small $\gamma$ is, for large enough $N$ we can choose a set of size $< \gamma N$ that contains $\hat{S}$. This set has $|\delta(S)| \geq |\delta(\hat{S})|$ and $|N(S)| = |N(\hat{S})|$ (since $\hat{S}$ is connected to every constraint). Consequently, for all $\gamma > 0$, $|N(S)|/|\delta(S)| \to 0$ as $k \to \infty$.

# S7 Sparse higher-order Hopfield networks generically have robust exponential capacity

In the next section, we will construct Hopfield networks with pairwise connectivity that implement the dynamics of the error-correcting codes defined in the previous section. Before doing this, we return to the results of Section S3. Recall that in Section S3 we showed that the codewords of a general linear code could be mapped to the energy minima of a higher-order Hopfield network, but that the error-correction did not correspond to the Hopfield dynamics. Thus, while these networks have exponentially-many minima, they do not have large basins of attraction (i.e., growing as $\sim N$) surrounding these minima. We now use this mapping and insights from expander codes to show two things. First, higher-order Hopfield networks can implement an expander code (including the error-correcting dynamics) and thus can have both exponentially-many minima and large basins of attraction. Second, this result is generic, in that it is true of a random higher-order Hopfield network, provided that it satisfies the mild conditions required for the equivalent random bipartite graph to be an expander (see Lemma 1 for example). Note that in this section we consider Hopfield networks on $\{-1, +1\}$, for convenience, but that given such a network there is an equivalent network on $\{0, +1\}$ with a slightly more complex energy function.

**Theorem 1.** *Higher-order Hopfield networks of order $z_C$ on $N$ neurons and with unit weight connections can implement an expander code on $N$ variables with constraints that each contain $z_C$ variables.*

*Proof.* Consider an expander code that corresponds to a bipartite graph with $N$ variables, $N_C$ parity check nodes, and left and right degrees $z$ and $z_C$ respectively. Thus the codewords of this code are defined by the set of constraints $\sum_{i \in C_i} x_i = 0$, where each constraint $C_i$ is a sum of $z_C$ terms. Consider the Hopfield network whose neurons, $s_i$ take states in $\{-1, +1\}$, with energy function $E(\boldsymbol{s}) = -\sum_{C_i} \prod_{i \in C_i} s_i$. By Claim 1, the minima of this energy function correspond to the codewords of the expander code.

Now consider the bit flip error-correction algorithm described in the previous section. Consider a variable node $x_i$, which participates in $S$ satisfied and $U$ unsatisfied constraints. Under bit flip dynamics, this node changes state if $U > S$. In the equivalent Hopfield network, this is exactly the condition that flipping the neuron reduces energy. Thus, a simple unit weight higher-order Hopfield network can implement bit flip dynamics. Note that this argument can easily be extended to variable degrees at the neurons and constraints. $\square$

We next show that this result is true of randomly-connected networks.

**Theorem 2.** *Consider a randomly-connected higher-order Hopfield network of order $z_C$ on $N$ neurons, where each neuron participates in $z$ edges, with unit weight. Assume that $z \geq 5$ and $z_C \leq C$ for some $C$. Then, with probability approaching 1, for some $\alpha, \beta > 0$, this network has $2^{\alpha N}$ stable states and can recover each stable state from an initial condition perturbed on $\beta N$ neurons.*

*Proof.* Consider a bipartite graph defined by this higher-order Hopfield network, where each of the $N$ left nodes corresponds to a neuron, each of the $N_C$ right nodes corresponds to an interaction term (i.e., one of the higher-order constraints), and a left and right node are connected by an edge if the corresponding neuron participates in the corresponding constraint. If the edges in this graph are chosen randomly, then by Lemma 1[1;16] this graph is an expander. Consequently there is an expander code defined on this bipartite graph. By Theorem 1, the higher-order Hopfield network implements this expander code and hence has exponential capacity (i.e., number of stable states $2^{\alpha N}$ for some $\alpha > 0$) and can recover these states from a finite fraction of perturbed variables ($\beta N$ perturbations). □

The above proof is described for a certain fixed-degree expander construction, but the mapping is quite general and easily extended to variable degree and to non-random constructions.

Randomly-connected higher-order Hopfield networks are studied in physics in the context of spin glasses (often under the name $p$-spin infinite range models)[17]. The isomorphism between these models and expander codes that we have identified remains to be further explored and may be a source of new ideas and techniques for understanding spin glass models.

# S8 Hopfield network expander codes: construction

We now construct regular Hopfield networks (i.e., pairwise connectivity) that implement the dynamics of the error-correcting codes defined in the previous section. Note that the construction we present is simply one possible implementation of a broader idea. In this section we first define a single, isolated *constraint node*, a small Hopfield network that will serve as a building block of our construction. We then describe the connections of the full network and show that it has exponentially-many energy minima. In the section after, we describe the memory retrieval dynamics of this network.

## S8.1 Constraint nodes

**Definition 2.** *Constraint node. Consider a Hopfield network with a set of $z_C$ neurons designated as "input" neurons, and $K$ neurons designated as "constraint" neurons. The $z_C$ input neurons can take $2^{z_C}$ possible states, and we will designate some subset $S = \{\mathbf{v}_1, \ldots, \mathbf{v}_K\}$ of these to be "preferred" states, where preferred states differ on the value of at least two input neurons. Note that the number of preferred states equals the number of constraint neurons $K$. Let $U_{ij}$ be the connection between the $i$-th input neuron and the $j$th constraint neuron.*

We set $U_{ij} = 1$ if $v_j(i) = 1$ and $U_{ij} = -1$ if $v_j(i) = 0$, where $v_j(i)$ is the state of the ith input neuron in the j-th preferred state. These connections will cause the jth constraint neuron to prefer the jth pattern. In its preferred state, the jth constraint neuron will receive input $\sum_i v_j(i)$ (i.e., the number of nonzero entries in its input pattern). To ensure that all constraint neurons receive the same amount of input in their preferred state, we also add a fixed bias of $b_j = z_C - \sum_i v_j(i)$ to the jth constraint neuron. Finally, we add inhibitory connections of strength $z_C - 1$ between all constraint neurons.

A schematic of the above construction is in Fig. 3a,b in the main text. Note that if $\mathbf{x}$ is the state of the input neurons and $\mathbf{h}$ the state of the constraint neurons, then the above network has energy function

$$E(\mathbf{x}, \mathbf{h}) = -\left( \mathbf{x}^T U \mathbf{h} + \mathbf{b}^T \mathbf{h} + \frac{1}{2} \mathbf{h}^T W \mathbf{h} \right), \tag{11}$$

where $U$ and $b$ are defined in Definition 2, and $W$ is the matrix with diagonal elements set to 0 and off-diagonal elements set to $-(z_C - 1)$.

**Claim 3.** *The energy minima of the network constructed in Definition 2 correspond to states with $\mathbf{x} = \mathbf{v}_j$, $h(j) = 1$ and $h(k \neq j) = 0$, for some j. Further, if the inputs are clamped to a preferred state $\mathbf{x} = \mathbf{v}_j$ then, regardless of the initial state of the constraint neurons, the constraint node settles into a state where $h(j) = 1$ and all other neurons are 0.*

*Proof.* First, note that the states described above are local minima. If the network is in the above state, then changing any element $x(i)$ increases the value of the energy term $-x(i)h(j)$ from $-1$ to $+1$ while leaving all other terms fixed. Setting $h(j) = 0$ increases the energy $-\sum_i x(i)h(j)$ from $-z_C$ to 0. Finally, setting $h(k \neq j) = 1$ increases the energy $-h(k)h(j)$ from 0 to $z_C - 1$ while at most decreasing the energy of $-x(i)h(k)$ from 0 to $z_C - 2$ (recall that the stable state preferred by constraint neuron $k$, $\mathbf{v}_k$, differs from $v_j$ on at least 2 input neurons). Thus the state is a local minimum.

Second, no other state is a local minimum. To see these, we enumerate states by the number of active constraint neurons.

1. No constraint neurons active: in this case the input neurons are unconstrained and can freely change state.

2. One constraint neuron active, call this $h_j$: If the input neurons are not in $\mathbf{v}_j$, it is energetically favorable to change them in the direction of $\mathbf{v}_j$.

3. Two constraint neurons active, call them $h_j$ and $h_k$: There is at least one input neuron in the state preferred by $h_j$ but not $h_k$ or vice versa. Changing the state of this neuron does not increase the energy.

4. More than two constraints neurons active: Turning off one of these neurons decreases the energy term $\frac{1}{2}\mathbf{h}^T W \mathbf{h}$ by at least $2(z_C - 1)$, and increases the remaining terms by at most $z_C$. Since $z_C \geq 2$ this decreases energy.

Next, consider the case where the inputs are clamped to a preferred state $\mathbf{v}_j$, so that $\mathbf{x} = \mathbf{v}_j$. If $||\mathbf{h}||_1 = 0$ or $||\mathbf{h}||_1 = 1$ but $h(j) = 0$ then it is energetically favorable for $h(j)$ to turn on. If $||\mathbf{h}||_1 \geq 2$ then there is some active constraint neuron $\mathbf{h}(k) = 1$, for $k \neq j$. Turning off this neuron will increase the input energy by at most $z_C - 2$ while decreasing the energy from recurrent inhibition by at least $z_C - 1$ and is thus energetically favorable. Hence the only stable state has $||\mathbf{h}||_1 = 1$ with $\mathbf{h}(j) = 1$. $\qquad\square$

Thus, as a consequence of the strong inhibition, the constraint network has competitive dynamics: in the lowest energy state the input neurons are in a preferred configuration, the neuron in the constraint network corresponding to this configuration is active, and all other neurons are suppressed. Note that the recurrent inhibition can be replaced by non-specific strong background inhibition to all constraint neurons. This change does not sacrifice accuracy of the final result, but it slows down network convergence by a constant factor (independent of $N$).

We next consider the dynamics of the constraint node when its input neurons are clamped to a state with Hamming distance of 1 from a preferred state.

**Claim 4.** *Assume that the input neurons to a constraint node are clamped to a state $\mathbf{x} \neq \mathbf{v}_j$ for all $j$ but $||\mathbf{x} - \mathbf{v}_j|| = 1$ for at least one $\mathbf{v}_j$. Consider the set $W = \{\mathbf{w}_1, \ldots, \mathbf{w}_L\}$ of states with Hamming distance 1 from $\mathbf{x}$ (this set at least contains $\mathbf{v}_j$). The network wanders through all states in which either one or two of the constraint neurons that prefer states in $W$ are active and all others are 0.*

*Proof.* The proof proceeds by demonstrating that the states described above are the only stable states. Regardless of the input state, it is never energetically favorable for more than two constraint neurons to be active, and so we will only consider states where 0 to 2 constraint neurons are active. Consider the set of constraint neurons preferring states from the set $W$ of states with Hamming distance 1 from $\mathbf{x}$.

1. If $||\mathbf{h}||_1 = 0$ or $||\mathbf{h}||_1 = 1$ but the active neuron does not prefer a state in $W$, then activating a constraint neuron that prefers a state in $W$ does not increase energy and is permitted.

2. If $||\mathbf{h}||_1 = 2$ but one of the active constraint neurons does not prefer a state in $W$, then turning off this neuron will increase the input energy by at most $z_C - 2$ while decreasing the energy from recurrent inhibition by $z_C - 1$ and is thus energetically favorable.

3. Thus the allowable states are where either $||\mathbf{h}||_1 = 1$ or $||\mathbf{h}||_1 = 2$ and the active constraint neurons prefer states in $W$. These are energetically equivalent and thus the network can (and does, given the stochastic nature of sequential updating in the Hopfield network) wander between all possible combinations.

$\qquad\square$

Finally, we make an observation about when input neurons are able to change state for a single isolated constraint node.

**Claim 5.** *Consider an isolated constraint node whose inputs are not connected to any other constraint nodes. Assume that the input neurons to the constraint node are in a non-preferred state* $\mathbf{x}$, *and that* $d(\mathbf{x}, \mathbf{v}_j) = 1$ *for some preferred state* $\mathbf{v}_j$. *Let* $x(i)$ *be the neuron on which* $\mathbf{x}$ *and* $\mathbf{v}_j$ *differ. Assume that the constraint neuron corresponding to* $\mathbf{v}_j$, $h(j)$, *is active, either alone or in combination with another constraint neuron* $h(k)$ *that also prefers a state adjacent to* $\mathbf{x}$. *Then it is energetically favorable for neuron* $x(i)$ *to change state.*

*Proof.* Since $h(j)$ is active, switching the input state to $\mathbf{v}_j$ decreases the energy due to the interaction with $h(j)$ by one unit. If some $h(k)$ is also active, then the energy due to the interaction with $h(k)$ increases by one unit. Thus it is either energetically favorable or neutral to change the state of neuron $x(i)$. □

## S8.2 Hopfield networks with exponentially-many well-separated minima

We next use the constraint nodes constructed above to construct Hopfield networks with exponentially-many well-separated minima. Our construction relies on mapping expander codes to bipartite Hopfield networks, with variable nodes corresponding to input neurons, and constraint nodes corresponding to the Hopfield network constraint nodes described in the previous section. Note that in order for the capacity of this network to be truly exponential in all participating neurons, the number of neurons in each constraint node must not grow with the size of the network. For our construction above, this is true when the constraints are sparse, meaning that each constraint constrains a fixed number of inputs (as is true for LDPC codes in general and expander codes in particular).

**Theorem 3.** *Hopfield networks of* $N$ *neurons with pairwise connectivity can possess exponentially-many stable states corresponding to the codewords of an expander code.*

*Proof.* As in the section on expander codes, the networks we consider are bipartite, containing $N$ input neurons, which determine the states or memories that will be stored and corrected, and $N_C$ constraint nodes, which determine the allowed states of variables they are connected to (see Figure 3). However, now these constraint nodes are themselves small networks of neurons, which have been described above. The $i$th input neuron connects to $z^{(i)}$ constraint nodes, and the $j$th constraint node connects to $z_C^{(j)}$ inputs. Consequently $\sum_i z^{(i)} = \sum_j z_C^{(j)}$ (i.e., the number of edges leaving the input neurons equals the number of edges entering the constraint nodes). $z$ and $z_C$ are small and chosen from distributions that do not scale with $N$; consequently the networks are sparse. By Lemma 1, these networks are $(\gamma, (1 - \epsilon))$ expanders with $\epsilon < 1/4$ for appropriate choices of $z, z_C$.

There are $z_C$ variables connected to a given constraint node and these could take any of $2^{z_C}$ possible states. The constraint nodes restrict this range, so that a subset of these states have low energy (and are thus preferred by the network). While there are multiple possible ways to construct constraint nodes, in the constructions we show each neuron in a constraint node prefers one possible configuration of the input neurons (see Definition 2 for details).

The Lyapunov (generalized energy) function of the network is:

$$E(\mathbf{x}, \mathbf{h}) = -\left( \mathbf{x}^T U \mathbf{h} + \mathbf{b}^T \mathbf{h} + \frac{1}{2} \mathbf{h}^T W \mathbf{h} \right) \tag{12}$$

here $\mathbf{x}, \mathbf{h}$ are the activations of the input neurons and the neurons across all constraint nodes, respectively; $\mathbf{b}$ are biases in the constraint neurons; $U$ are the symmetric weights between input and constraint neurons (as defined previously in the section on constraint nodes); and $W$ are the lateral inhibitory interactions between neurons within the constraint nodes.

The network is in a stable or minimum energy state when the input neurons are in a state that satisfies all of the constraint nodes, and when the constraint neurons preferring the corresponding input states are active.

If each constraint is satisfied by a fraction $2^{-r_j}$ of possible states, where $r_j \geq 1$ but is not necessarily integer, then the average number of minimum energy states for the network is

$$N_{\text{states}} = 2^{-\langle r \rangle N_C} 2^N = 2^{N - \langle r \rangle N_C} = 2^{\left(1 - \langle r \rangle \frac{\langle z \rangle}{\langle z_C \rangle}\right) N}. \tag{13}$$

Here the angle brackets represent averages, and the last equality follows because $N\langle z \rangle = N_C \langle z_C \rangle$. If the constraints are defined to correspond to parity equations then this result is exact (with $r_j = 1$, as in Fig. 2). For general constraints this result only holds true in expectation, under the assumption that the constraints independently restrict the set of possible states (see S12 for further discussion of the case when acceptable states for each constraint are chosen randomly, and Fig. S3 for numerical verification of the mean-field argument in this case).

For notational convenience define $\hat{z} = \frac{\langle z \rangle}{\langle z_C \rangle}$. Thus, as long as $\langle r \rangle \hat{z} < 1$, the expected number of stable states grows exponentially with $N$. However, $N$ is the number of input neurons and not the total network size. Each constraint network has at most $K_{max} = 2^{z_C^{max} - 1}$ neurons, and thus the total number of neurons in the network is at most $N_{total} \leq N + K_{max} N_C = (1 + K_{max}\hat{z})N$. Since $z$ and $z_C$ are drawn from fixed distributions, this prefactor does not grow with network size. The number of minimum energy states is

$$N_{\text{states}} \geq 2^{\alpha N_{total}} \quad \text{where} \quad \alpha = (1 - \langle r \rangle \hat{z})/(1 + K_{max}\hat{z}) \tag{14}$$

Thus the expectation of the number of minimum energy states grows exponentially in the total size of the network. $\qquad\square$

For the simulations in Figure 2, we set $N_C = 0.95N$, and choose $Z = 4 + Z_{add}$, where $P(Z_{add} = k) = 0.85 \times 0.15^{k-1}$ (note this is a geometric distribution with $p = 0.85$). We then randomly assign outgoing edges from input neurons to constraint nodes, subject to $2 \leq Z_C \leq 6$. In Figure S3d,e, we choose deterministic values of $z = 5$ and $z_C = 12$ (and consequently $N_C = 5N/12$).

For Figure 2 we choose the preferred configurations of the constraint nodes to be parity states of the connected input neurons (this simplifies the numerical simulations, but is not necessary), and in Figure S3d,e we choose these configurations randomly, but subject to the constraint that each preferred configuration differs from the others in the state of at least two neurons.

# S9 Hopfield network expander codes: error-correcting dynamics

In this section we show that the Hopfield network dynamics carries out the decoding algorithm[1] described in Section S6. We first proceed under the assumption of a separation of timescales, where constraint neurons change much more rapidly than input neurons. We then show that this separation of timescales can be made arbitrarily large and thus the approximation holds to any degree required.

**Theorem 4.** *The dynamics of the Hopfield networks defined in Theorem 3 implement the bit-flip decoding algorithm described in Section S6.*

*Proof.* We first proceed under the assumption that the rate of changes in constraint neurons is much more rapid than in the input neurons (explained below). Thus, if the constraint neurons are not in equilibrium with the input neurons, first consider the case where the input neurons remain in some fixed state while the constraint neurons settle into equilibrium with that state.

First consider a constraint node with fixed inputs. If the input corresponds to one of the preferred states of the constraint node, $\mathbf{v}_j$, then as in Claim 3, this node settles to a WTA state where $h(j) = 1$ for the appropriate constraint neuron, and all the rest are inactive. If the input does not correspond to a preferred state for the node, there are neurons that prefer a neighboring state and these are weakly driven; either one or two of the weakly-driven neurons can be active, and the node drifts between all 1-sparse or 2-sparse activity combinations of the weakly-preferred neurons (Claim 4).

We next identify the conditions under which the $i$th input neuron will change its state. Define the current state of the input pattern to be $x$ and the state of the input pattern with input neuron $i$ flipped to be $x_{\neg i}$. Assume that the $i$th neuron is connected to a set of constraint nodes, $C$. We divide $C$ into three groups, which we call $C_S$, $C_U$ and $C_{U'}$. The constraint nodes in $C_S$ are in a WTA state with each one's active neuron's preferred input matching the current input. The constraint nodes in $C_U$ are the ones for whom neither $x$ nor $x_{\neg i}$ corresponds to any preferred state. The nodes in $C_{U'}$ do not have any preferred states matching $x$ but have preferred states that would match $x_{\neg i}$. The internal states of the $C_{U'}$ nodes move between the 1- and 2-sparse combinations of activity of neurons that prefer neighboring states of $x$ (Claim 4). Within one constraint node of $C_{U'}$, some of these neighboring states prefer a flip in $i$, and others do not. At a given time $t$, a fluctuating number $M(t)$ of the constraint nodes in $C_{U'}$ are in a neighboring state that prefers a flip in input $i$, with $0 \leq M(t) \leq |C_{U'}|$. When $M(t) > |C_S| + |C_U|$ the input neuron $i$ is energetically favored to flip its state according to the Hopfield dynamics, and will do so. Recall that in the bit-flip algorithm, the $i$th input flips whenever $|C_{U'}| > |C_S| + |C_U|$.

The number of internal states of the constraint nodes is $< 2^{z_C - 1}$ and remains fixed with $N$. Thus the number of neighboring states for unsatisfied nodes also does not scale with $N$, and the time taken for $M(t)$ to reach $C_{U'}$ (or any value $> |C_S| + |C_U|$ if $|C_{U'}| > |C_S| + |C_U|$) is finite and does not change with $N$. Thus, it follows that in the Hopfield network, the $i$th input will flip with finite probability when doing so is favorable according to the bit-flip algorithm, and will not do so otherwise. Therefore, the network implements the bit-flip

dynamics of Section S6 and, consequently, can correct a number of errors that scales with network size. □

**Corollary 1.** *The basins of attraction of the embedded stable states in the Hopfield network grow linearly with with network size, and the time taken for the network to converge is proportional to the number of errors and is hence $O(N)$.*

*Proof.* We have shown that the Hopfield network of Section S8 implements the bit flip decoding of the expander codes in Section S6, thus the basins of attraction map onto the basins in Section S6, and are linear in network size.

Each time an input neuron updates its state, the number of constraint nodes in high-energy (unsatisfied) states decreases by at least one. The initial number of errors in the input pattern is proportional to network size (O(N)). The maximal number of initially unsatisfied/high-energy constraint nodes is $z_{max}$ times the number of errors and thus also $O(N)$. Consequently, the time to converge is on the order of the product of the time to update an input neuron with the number of errors.

The update of an input neuron depends on coincident activation of constraint neurons across a fraction of its connected constraint nodes ($O(z)$) that prefer the flipped state. The time taken for this update to happen scales with $z, z_C$, but not $N$, because $z, z_C$ are independent of $N$. Thus, the number of steps taken for the Hopfield network to converge is also $O(N)$, like the bit-flip algorithm, albeit with a larger prefactor. □

The proofs above rely on the assumption that the rate of changes in constraint neurons is high enough that their states are in equilibrium with the states of the input neurons. This assumption holds because an input neuron must wait to switch until enough of its unsatisfied constraint nodes, through explorations of combinations of the neighboring states, push it into the new, lower-energy state. By contrast, when an input neuron switches states, the constraint neurons with matching preferred input in a previously unsatisfied constraint are either already active (this was a condition for causing the input neuron's switch) or activate as soon as it is their turn to be updated, and the constraint node state immediately reflects the low-energy configuration for the given input state.

Also note that the separation of timescales between input and constraint neurons can be made arbitrarily large by choosing the degree of the input neurons to be some sufficiently large but fixed value (that does not grow with $N$, so that the network remains sparse). Thus the separate timescales can be made arbitrarily large, if desired for theoretical reasons.

Finally note that like most codes, these Hopfield networks are not perfect codes, meaning that the codewords (i.e, the desired stable states) and the surrounding points that map to them (i.e., the basins of attraction) do not occupy the entire space of possible messages. Indeed, in high dimensions, the majority of the state space lies in between these spheres and the network has a large number of shallow local minima in the spaces between the coding spheres.

# S10   Weakening constraints

In the previous section we considered constraints with a guaranteed minimum distance of 2 between their satisfied states. We now extend those results to consider weaker constraints

that accept some fraction of adjacent states. For example, such a constraint might be satisfied when its connected variables take configurations $(0, 0, 0)$, $(0, 0, 1)$, and $(1, 1, 1)$. We define $p$ to be the probability that a neighbor of a satisfied state is unsatisfied. We show that for slightly higher expansion, the results above will hold on average, and we bound the deviation from this average for large $N$.

**Claim 6.** *Consider a Hopfield network expander code on a $(\gamma, (1 - \epsilon)$ expander graph, where constraint nodes are satisfied by neighboring states with probability $1 - p$. If $\frac{1}{2(1-2\epsilon)} < p$, then this network has exponential capacity and can show robust error correction.*

*Proof.* As in section S6, consider a state $Q$ differing from a satisfied state $Q_{sat}$ on a set of neurons $E$. In section S6 we had Unique$(E) \subset U(E)$ (and, implicitly, $p = 1$). However, in the weaker setting, a constraint in Unique$(E)$ is unsatisfied with probability $p$. For notational convenience define the random variable $X = |U(E)|$ (i.e., the number of unsatisfied constraints), and note that $X \geq$ Binomial$(K, p)$, where $K = |$Unique$(E)| \geq (1 - 2\epsilon)|\delta(E)|$. As before, we wish $X > |\delta(E)|/2$.

For this to hold we will require higher expansion. Choose $\epsilon$ so that $\frac{1}{2(1-2\epsilon)} < p$ (if $p = 1$ then $\epsilon < 1/4$ as before). Now define $\alpha = \frac{1}{2p(1-2\epsilon)} < 1$ and note that $\mathbb{E}[X] \geq pK > |\delta(E)|/(2\alpha) > |\delta(E)|/2$. Thus, on average, the state $Q$ will have a neuron that it is energetically favorable to flip.

We now bound the probability of error, meaning the probability that the number of unsatisfied constraints is less than half the number of edges leaving $E$.

$$
\begin{aligned}
P\left(X < \frac{|\delta(E)|}{2}\right) &\leq P\big(X < \alpha\mathbb{E}[X]\big) \\
&\leq \exp\left(-\frac{(1-\alpha)^2}{2}\mathbb{E}[X]\right) \\
&< \exp\left(-\frac{(1-\alpha)^2}{4\alpha}|\delta(E)|\right).
\end{aligned}
\tag{15}
$$

Here we use the Chernoff bound for the second inequality.

$|\delta(E)|$ is approximately proportional to the number of error variables, $|E|$ (for a fixed degree network this is exact and the proportionality constant is just the degree). Thus, the probability that there exists another minimum within a distance $d$ of $Q_{sat}$ falls off exponentially in $d$.

These results allow the presence of other local minima, which we can divide into two categories. First, there may be a small number of local minima very close to $Q_{sat}$, at a distance that does not scale with network size (and thus a distance that vanishes in relative terms). The effect of these minima is to slightly expand the desired energy minima to possibly include a set of nearby states rather than a single state, but the size of this set does not grow with network size or number of minima. Second, while the probability that there exists another minimum within a distance $d$ of $Q_{sat}$ falls off exponentially in $d$, the number of states at distance $d$ grows exponentially in $d$, and thus there will be $O(1)$ local minima at distance $d$. However, since these minima are produced by rare events, the basins of attraction are likely to be small and most trajectories should not see these minima.

Next, we note that the probability of these local minima decreases exponentially in the variable degree. To see this, consider Eq. 15 for a fixed degree network, where each variable participates in $z$ constraints. Then $|\delta(E)| = z|E|$,

$$P(\text{local minimum}) < \exp\left(-\frac{(1-\alpha)^2}{4\alpha}|\delta(E)|\right)$$

$$= \exp\left(-\frac{(1-\alpha)^2}{4\alpha}z|E|\right). \tag{16}$$

Thus it is always possible to achieve an error below any given fixed probability by choosing $z$ appropriately and this value of $z$ does not need to grow with network size. Moreover, the error probability can be asymptotically driven to 0 by allowing $z$ to grow with $N$ at any rate. $\qquad\square$

# S11    Noisy updates / finite temperature

Figure S2: **Network dynamics at finite temperature.** (a) Fraction of times the network infers the correct state as a function of percent input corrupted, for two different inverse temperatures. The network is considered to have reached the correct state if the final state is within two standard deviations of the mean of the equilibrium distribution shown in **b**. (b) Distribution of number of nonzero variable neurons for networks started at the all-zero energy minimum and allowed to evolve. Note that hidden neurons are subject to noisy updates as well (state not shown). Colors correspond to **a**. (c) Sample trajectories for two values of inverse temperature. Top panel shows inverse temperature $= 4.0$ and bottom panel shows inverse temperature $= 4.5$. Network size of 500 variables in all simulations.

Previously, we established the existence of an energy gradient allowing error correction with large basins of attraction. We now consider the case when input neurons update their state probabilistically rather than always descending the energy gradient. We show that the network state evolves towards the energy minimum on average and, as before, we bound the deviation from this average for large $N$. Note that in this case we do not expect perfect decoding. If we start the network at an energy minimum and there is some small probability $p$ of a neuron flipping to a higher energy state, then on average $pN$ neurons will flip, and thus the distribution of network states will be localized around, but not exactly at, the minimum.

**Claim 7.** *Assume that neurons update their state probabilistically, with some probability $p$ to switch to a higher energy state, where $p < \gamma/2$ ($\gamma$ is the size of sets that expand). Consider a network state within a distance $\alpha N$ of an energy minimum, where $p < \alpha < \gamma/2$. The network is driven towards the energy minimum with probability $\exp\left(-\frac{(\alpha-p)^2 \phi N}{2}\right)$.*

*Proof.* Consider a neuron deciding which state to take, with $Q$ and $Q^\sharp$ the possible network states (differing only on the value of that neuron). Assume that $P(Q) = 1 - P(Q^\sharp) = f(\Delta E(Q, Q^\sharp))$, where $\Delta E$ is the energy difference between the two states and $f$ is some function. For a Hopfield network this function always picks the lower energy state, while for a Boltzmann machine the ratio of probabilities for the two states is exponential in the energy difference.

We consider the basin of attraction around a minimum energy state $Q_{sat}$ and consider some general state $Q$ in this basin of attraction. The energy of state $Q$, $E(Q)$ is proportional to $N_{VC}$, the number of violated constraints, with some constant $k_1$ (the constant is irrelevant for the Hopfield network formulation but not when the switching probability depends on the energy difference, such as with a Boltzmann machine). If $F$ is the set of error locations, then the previous analyses show that the number of violated constraints, $N_{VC} \geq (1 - 2\epsilon)|\delta(F)|$. Moreover, no constraint is violated unless it receives at least one edge from a variable in $F$. Thus $N_{VC} \leq |\delta(F)|$. For simplicity, we consider the case when the degrees are constant, so that $|\delta(F)| = z|F|$ (but note that we're considering sets whose size scales with $N$, and thus $|\delta(F)|$ will be increasingly concentrated around $\langle z \rangle |F|$ in the large $N$ limit). Combining,

$$k_1(1 - 2\epsilon)z|F| \leq E(Q) \leq k_1 z|F| \tag{17}$$

We make the approximation that $E(Q) \propto |F|$. Thus increasing the size of $|F|$ by 1 changes the energy by some fixed value $\Delta(E) > 0$ and decreasing $|F|$ changes energy by $-\Delta(E)$. Consequently, each neuron in $Q$ takes the same state as it does in $Q_{sat}$ with probability $1 - p$ and takes opposite state with probability $p$, where $p$ is small and depends on $\Delta(E)$.

In equilibrium, the average size of $F$ will be $pN$ and, for large $N$, fluctuations around this will be on the order of $\sqrt{N}$. Recall that the basins of attraction have size $\geq \frac{\gamma N}{2}$. Thus in order for decoding to work we require that $p < \frac{\gamma}{2}$.

We analyze the effect of updating a state $Q$ where $|F| = \alpha Q$, meaning that a fraction $\alpha$ of neurons have a state different from the energy minimum. We assume $p < \alpha$, since $pN$ is the best decoding possible and that $\alpha < \frac{\gamma}{2}$, so as to keep the state within the basin of attraction. We show that updates send the network towards state $Q_{sat}$ with high probability.

Consider the network after $M = \phi_{update}N$ neurons have been updated. If neurons update their states in parallel, as typically assumed, then this happens in constant time; if they

instead update sequentially then we are effectively considering some constant fraction of the time the network takes to converge. Let the random variable $X_i^{old}$ take value 1 if the $i$-th neuron in the update set is in error (i.e. differs from its state in $Q_{sat}$) and 0 otherwise. Similarly, $X_i^{new}$ is the corresponding random variable after the update. Initially the number of error neurons in this set is $X^{old} = \sum_{i=1}^{M} X_i^{old} \sim Bin(M, \alpha)$. After the update the number of error neurons in this set is $X^{new} = \sum_{i=1}^{M} X_i^{new} \sim Bin(M, p)$. Consequently, the set $|F|$ changes in size by $\Delta F = X_{error,new} - X_{error,old}$, and we wish to show $P(\Delta F < 0)$ vanishes as $N$ gets large.

Note that $\Delta F = \sum_i Y_i$, where $Y_i = X_i^{new} - X_i^{old}$. Also, $\mathbb{E}[\Delta F] = (p - \alpha)N$. Applying Hoeffding's inequality we find

$$
\begin{aligned}
P(\Delta F \geq 0) &= P\left( (\Delta F - \mathbb{E}[\Delta F]) > \frac{(\alpha - p)}{N} \right) \\
&\leq \exp\left( -\frac{(\alpha - p)^2 \phi N}{2} \right) \quad (18)
\end{aligned}
$$

Thus this probability decreases exponentially in $N$. Note that the average step size $\Delta F$ gets smaller as $\alpha \to p$. $\qquad\square$

For the simulations shown in Fig. S2, we scale all the connections described in Section 6 by an inverse temperature $\beta$, and update neurons according to Boltzmann dynamics (Eq. 3).

# S12    Self-organization to exponential capacity

We have showed that if the preferred patterns at each constraint node differ by at least two bits, the network as a whole has both exponentially many minima and large ($\sim N$) basins of attraction. Here, we show how the neurons within each constraint node, connected to the same subset of input neurons, can come to prefer sufficiently non-overlapping patterns.

## S12.1    Learning rule

We first describe the learning at a single constraint node. The neurons in the constraint node receive input from the same set of $z_C$ input neurons, with weak and non-specific initial weights (Fig. S3a). Assume that the constraint node has $K$ neurons, where $K \geq 2^{z_C - 1}$. Each of these $K$ constraint neurons inhibits the others with recurrent inhibition of strength $\xi$ and receives background inhibition of strength $\eta$.

On each learning step, we first provide a random input to the $z_C$ input neurons and allow the constraint neurons to reach equilibrium with the inputs. If the input is within a Hamming distance of 1 from a previously learned state then, provided $|\eta| < (z_C - 1)$, the constraint neuron corresponding to that state activates, otherwise all constraint neurons remain inactive. We then provide an excitatory input of strength $\zeta > |\eta|$ to a randomly selected constraint neuron. If no other constraint neuron is active (i.e., the state has not been learned before), then the neuron that receives this excitatory input activates, and we learn connection strengths of $+1$ with input neurons that are active and $-1$ with input

Figure S3: **Input-driven self-organization of weights in a network with random connectivity.** (a) Schematic of weight updating at a single constraint node. Initially the constraint node receives weak, non-specific projections from a subset of input nodes (first panel). Hebbian plasticity then associates random input patterns with sparse random activation of constraint neurons. If the input to a constraint node is not close to a previously seen pattern that has become a preferred pattern through plasticity, and if a constraint neuron in the node is active, the active constraint neuron learns connection strengths that prefer this pattern (second and third panels; fifth and sixth panels). If the constraint node receives an input that is close to a previously learned pattern, then the constraint neuron corresponding to this pattern activates and suppresses the activation of other constraint neurons (fourth panel). (b) Mean time taken for self-organization as a function of number of constraint nodes (black circles) along with logarithmic fit (gray line). Time grows logarithmically with network size. (c) Number of global minima when preferred states at a constraint are chosen randomly (subject to minimum separation of 2 in Hamming distance), shown for small networks. Blue circles show mean number of global minima for networks with input degree ($z$) of 3 and constraint degree ($z_C$) of 8. Blue line shows exponential scaling with exponent predicted from mean-field argument (Eq. 13). Magenta, red and black show results for other combinations of ($z, z_C$) (labeled on plots). (d) Number of fixed points for learned networks, predicted from a mean field analysis (explicit results for small networks shown in panel (c)). Black circles show individual points from the predicted distribution; gray line shows the mean. (e) Error-correction performance of network after self-organization with $N = 480$. Error bars show standard error.

neurons that are inactive, and we add a bias that ensures that the total input drive is $z_C$. If another constraint neuron is active (i.e., the state has been learned before) then, provided that $|\zeta| < |\xi| + |\eta|$, this constraint neuron suppresses the activation of other constraint neurons via the recurrent inhibition and no learning takes place. At the end of learning, we remove the inhibitory bias $\eta$. Parameters are $\xi = -(z_C - 1)$, $\eta = -(z_C - 1.5)$ and $\zeta = z_C - 1$.

In sum, the learning proceeds by pairing a very-sparse random drive of the constraint neurons (so that at most one constraint neuron is activated) with random activation of the inputs, and updating weights in a Hebbian-like one-shot modification. The background input to the constraint neurons ensures that all constraint neurons receive the same average input over time, regardless of their (learned) preferred input configurations.

Learning is considered complete when the constraint node has seen enough patterns to cover the input space, meaning that every one of the $2^{z_C}$ possible input patterns has either been learned by the constraint node or is adjacent (i.e., Hamming distance of 1) to a previously learned pattern. Note that the learning procedure just described is equivalent to randomly selecting satisfied constraint states from the $2^{z_C}$ possible inputs, subject to the constraint that each new selected state must be a Hamming distance of at least 2 from previously selected states.

For multiple constraint nodes, the learning described above proceeds in parallel at each node. Random patterns are presented to the $N$-neuron input layer of the network as a whole. Each constraint node receives input from a random subset of $z_C$ input neurons (determined by the random initial connectivity). Thus for each node learning proceeds using the values of the input patterns at the relevant $z_C$ input neurons and does not depend on the learning at any other constraint node.

## S12.2 Time taken for learning

First consider the time taken for learning at a single constraint node of degree $z_C$. Call this random variable $T_i$. As described above, input patterns are presented randomly and learning terminates when the constraint node has seen enough patterns to cover the input space, meaning that every one of the $2^{z_C}$ possible input patterns has either been learned by the constraint node or is adjacent to a previously learned pattern. Note that $z_C$ does not grow with network size, and thus the distribution of learning times at a single node is fixed with network size.

Let $\tilde{T}_i$ be the time taken for the network to see all possible input patterns and observe that $T_i$ is bounded by $\tilde{T}_i$, meaning that for any $t$, $P(T_i > t) \leq P(\tilde{T}_i > t)$. The problem of determining the time $\tilde{T}_i$ to see every one of $2^{z_C}$ input patterns is the well-known coupon collector problem[18]. In particular, $\mathbb{E}[\tilde{T}_i] \sim z_C 2^{z_C}$. $z_C$ is a small fixed number (and thus the mean learning time at a single node is constant).

Now consider the time taken to learn all $N_C$ constraints, which we call $T_{tot}(N_C)$. Observe that learning proceeds in parallel, and that the learning at each node does not depend on the rate of learning at any other node. The time for learning to finish is thus determined by $\max_i T_i \leq \max_i \tilde{T}_i$. For notational convenience define $\mu = z_C 2^{z_C}$. Then,

$$P(\max_i T_i > t\mu) \leq \sum_{i=1}^{N_C} P(T_i > t\mu) \leq \sum_{i=1}^{N_C} P(\tilde{T} > t\mu) = N_C P(\tilde{T} > t\mu)$$
$$\leq N_C 2^{-(t+1)z_C} = 2^{-(t+1)z_C + \log(N_C)},$$

where the final inequality is a simple tail bound for the coupon collector problem[18]. Now observe that setting $t = (1 + \epsilon)\log(N_C)/z_C$ suffices to ensure that $P(T_{tot} > t\mu) \to 0$ as $N_C \to \infty$. Thus, the learning time is $O(\log(N_C))$ with desired probability.

### S12.3 Numerical experiments

To confirm the learning time numerically, we simulate the learning process at a single constraint node of degree $z_C = 12$ to yield a distribution of learning times. We then use this empirical distribution to compute $T(N_C) = \mathbb{E}[\max_{1 \leq i \leq N_C} T_i]$. In Fig. S3b we plot $\log(T(N_C))$ as a function of $N_C$, verifying the logarithmic scaling.

For the numerical results shown in Fig. S3d,e, we start with a randomly-constructed bipartite graph, with degree 5 at the input neurons and 12 at the constraint nodes. Thus each input connects to 5 constraint nodes, and each constraint node constrains the state of 12 inputs. After learning is complete, each constraint is satisfied by a fraction $2^{-R}$ of its possible input states, where $R$ is a random variable determined by the learning procedure.

To estimate the capacities for the plot in Fig. S3d, we first numerically compute the distribution of $R$. Then for a given number of input nodes $N$ and constraint nodes $N_C$, we plot samples of $2^{N - \sum_{i=1}^{N_C} R_i}$ (i.e., Equation 13), where the $R_i$'s are independently drawn from the distribution for $R$.

This predicted capacity relies on the assumption that the constraints independently restrict the set of possible input states. We numerically verify this mean field argument in Fig. S3c, where we explicitly count the number of global energy minima for small networks of various sizes and degrees and plot the results along with the predicted exponential scaling, with exponent given by Eq. 13.

For Fig. S3e, we choose acceptable input states for a constraint node randomly, subject to a minimum Hamming distance of 2 between selected states (this procedure is equivalent to the learning procedure), and set the weights between the input and constraint neurons appropriately. For convenience, we choose the same set of preferred states for each constraint node (note that the results of S9 apply to any set of constraints as long as there is a minimum distance of 2 between allowed states at each constraint).

## S13 Notes on capacity results

Classical results in the theory of Hopfield networks show that a network of $N$ neurons cannot store more than $O(N)$ arbitrary patterns[8;9]. While these results are typically framed in the context of recurrent networks without hidden neurons, simple arguments show that the same results hold for more general architectures.

First, is it possible to circumvent bounds on the storage of arbitrary patterns through an alternate scheme, in which exponentially many arbitrary patterns are mapped to the robust memory states of a high-capacity network such as the one we construct? Encoders in communications theory do just this, mapping arbitrary inputs to well-separated states before transmission through a noisy channel. From a neural network perspective, the feedforward map can be viewed as a recurrent network with input and hidden units and asymmetric weights, so again we know from capacity results on non-symmetric weights[9] that it should not be possible. Mapping exponentially many arbitrary patterns to these structured memory states in a retrievable way would require specifying exponentially many pairings between inputs and structured memory states, and thus in general, equally many synapses. One way to obtain that many synapses would be to have exponentially many input neurons, but then the overall network would not possess exponential capacity for arbitrary many patterns as a function of network size.

Existing capacity results on Hopfield networks typically assume that all neurons are visible neurons. However, adding hidden neurons cannot change the scaling of capacity with network size. Consider a network of $N$ neurons with $N_1 = \alpha N$ visible neurons and $N_2 = (1 - \alpha)N$ hidden neurons. Assume that the network stores a set $S$ patterns on its visible neurons, meaning that for each state $s_i \in S$, there exists at least one stable state of the network dynamics where the visible neurons are in state $s_i$ (note that Hopfield networks always converge to a stable state). If the size of $S$ is $f(N_1)$, then there exists an equivalent network of $N = N_1 + N_2$ visible neurons (i.e., no hidden neurons) with at least $f(N_1)$ patterns, and thus the previous capacity results show that $f(N_1)$ is $O(N)$ (or $O(N^2)$ for sparse patterns).