[Reviews · NeurIPS 2019]

Reviewer 1



Regular Hopfield nets of N nodes can only store O(N) patterns. These patterns act as local minima to denoise inputs by moving them towards the stored patterns. The novel idea in this paper is to link this idea to that of ECCs. In ECCs, we have N bits of which M are the message and P are parity checksums, with N=M+P. (For example, M=7, P=3). ECCs have a similar effect to Hopfield nets in that the 2^M possible messages each have unqiue checksums, and act as attractors to noisy input versions. But unlike Hopfield, they have exponential (2^M) stored patterns rather than linear. This motivates an exploration of the question, how can ECCs have this larger capacity than Hopfields, and how can Hopfields be modified to be equavilent to ECCs. The first attempt uses modified Hopfield nets with 4-factors replacing pairwise factors. This can then be converted into an equivilent RBM-like bipartite structure with pairwise weights. I would have liked the links to RBM to be made more explicit -- mostly just as a cultural thing, most people in the community are familiar with RBMs due to the deep learning bandwagon so it's a standard concept to hang the new ideas off. e.g. Fig 1c has RBM structure. This first step is found not to work, because local minisiation doesn't always take us to the correct place. But another new idea, use of expander network toplogoes, is introduced, which encourages the minimisation to find the correct solutions. The paper is very well written and has clearly been through good internal review already. (Unusually I can't find any typos to report.) It's an interesting and novel paper which makes a new link between two previously unconnected areas and I think it could inspire more new thinking between them. Maybe it will do for Hopfield nets what Turbocodes did for Bayes nets. Accept.

Reviewer 2



Originality: The theoretical contributions are largely based on well-known ideas (Sipser and Spielman, 1996). However, the adaptation of these ideas to Hopfield nets is interesting and original. Significance: While associative memory networks are certainly interesting objects in their own right, as a submission in the neuroscience track, I could not say that I took away concrete neuroscientific insight from reading the paper. Indeed, the paper spends hardly any time on this except for a brief paragraph in the discussion. I feel like the paper will be much stronger with more discussion on relevance to the neuroscientific community. Quality: I did not check all the proofs in the supplementary material in detail, but the work appears to be technically sound and supported by simulations. Clarity: I feel like too much of the actual results and concrete details have been relegated to the appendix. For example, the main paper does not even include equations or much detail about the energy function, dynamics or learning of the proposed architecture. Oddly enough, the equations that are included are not of the main model. Minor: please include a legend in the panel in Figure 2c. ________ I've read the author's feedback. Thanks for your response and expanded discussion on relevance to neuroscience.

Reviewer 3



[As a general note to the meta-reviewer, the other reviewers, and the authors, this review was a last-minute emergency review and so unfortunately does not consider the extensive information in the supplementary] This work asks whether “it is possible to build simple, Hopfield-like neural networks that can represent exponentially many well-separated states using linearly many neurons, and perform error-correction or decoding on these states in response to errors on a finite fraction of all neurons.” As the authors note, capacity is often balanced against robustness; since code redundancy is needed to enable recovery from noise, capacity is necessarily reduced because of the redundancy, making this an especially difficult problem. The authors rise to this challenge and claim to produce a network that exhibits “exponential capacity, robustness to large errors, and self decoding or clean-up of these errors”. There network takes the structure of a restricted Boltzmann machine wherein the hidden units are comprised of clusters of neurons that laterally inhibit each other, each with the same connectivity to the input units but with different weights. The authors motivate their network using inspirations and ideas from expander graphs, error correcting codes, and Hopfield Networks. The proposed solution is straightforward and well-motivated, and all the design and algorithm choices seem quite sensible. Unfortunately, as a non-expert in this area (and given the short time available for review) I could not properly assess the significance or novelty of the approach in the context of the broader literature. On a general note, the manuscript is wonderfully written, and as a non-expert in this area it was easy to grasp the significance of the problem as well as the solution proposed. The authors are commended for putting together a clear, crisp, and easy to understand piece of work. Of particular note are the intuitions and insights scattered throughout the text, which help the reader grasp the logic of the proposed solution. I have the following questions that I hope the authors can comment on: - While acknowledging that this work is mainly conceptual, I am left wondering about the approach’s sensitivity to the type or form of data. My hunch is that the approach should be relatively robust, in which case I am wondering about the author’s decision to not experiment with other types of data (and potentially compare results to any relevant baselines, if they exist). - Building off the previous question, one constraint seems to be the requirement for binary inputs. Do the authors imagine a possible extension of this work to handle continuous valued data, or is this approach limited in this regard? - Can the authors provide further comments on the steep transition between error recovery and non-recovery as a function of noise (see figure 2C). Is there a reason for this fast transition, and/or would less severe slope be desirable?

[Author Response · NeurIPS 2019]

We thank the reviewers for their thoughtful comments.

**Proofs/technical content in main paper and SI (R1 and R3):** For R1, please note that a full set of technical proofs on capacity and decoding is in the SI of our original submission. As the reviewers can see, the proofs take up a lot of space, and so we decided to give a conceptual description in the paper, with specific pointers to the relevant proof in the SI for each of the results cited. In response to R3, we will add a paragraph in the main paper that outlines key equations/proofs, and makes it easier for the reader to locate the relevant proof in SI for a given result in main paper.

**Links to RBMs and to graphical models (R1):** Our network has RBM architecture, as noted by R1, if each constraint node is viewed as a unit (however, each constraint node is really a set of conventional neural units with recurrent connections within that set). We prove that the same exponential capacity and robust error correction extend to the stochastic update dynamics of Boltzman Machines (SI S11; pointer in line 274 of the original submission).

Our network can be represented as a factor graph (constraint modules are factors), undirected graphical model (clique potentials are indicator functions on visible neurons in a constraint) and Bayes net (in several different ways). None of these are the Bayes nets used by Mackay et al. to decode ECCs, but our network can be made structurally similar to these other Bayes nets by adding an input layer with fixed noisy inputs and slightly rewriting constraint modules (this is the LDPC Bayes net; turbo codes are not fundamentally different). The big difference is dynamical rather than structural. An expander graph code allows simple, neurally plausible decoding to perform at par with BP. The same simple decoding should apply to codes where the equivalent factor graph has an expander structure (e.g., random LDPC). These expander codes can also be decoded by belief propagation (BP), but it's harder the other way around. In general, codes that can be decoded by BP do not admit a simple neural network decoding by the dynamical/energy-based Hopfield rule: the message from node $i$ to $j$ depends on messages from all neighbors of $i$ except $j$; this leave-one-out structure is hard for biological neurons, though there are some interesting attempts. Thus, we do not expect general turbo code decoding to be performed by Hopfield dynamics. This is a fascinating research area and it will be interesting, in future work, to determine when the mapping is possible and to analyze concatenated codes as neural networks. We will add a short discussion of the relationship to Bayes nets/belief prop and slightly expand the discussion of RBMs.

**Relevance to neuroscience (R3):** We plan to follow this paper with another paper describing neuroscience applications. For space and coherence, this paper focuses on the conceptual theory without elaborating on applications. However, in response to these comments, we will add a few paragraphs discussing neuroscience applications, detailed below.

The networks we construct (henceforth HPC nets) can perform a number of canonical neural computations, with natural applications to neocortical-hippocampal interactions. The hippocampus plays an important role in neocortical memory, despite its puzzlingly small size ($\sim 10^7$ hippocampal vs. $\sim 10^{10}$ neocortical neurons in humans). Thus it is an appealing place to look for high-capacity networks. In particular, the HPC net can be used to construct a *robust high-capacity pattern labeler*. Here input patterns in a very high-dimensional space (putative neocortex) are mapped to the exponentially-many stable states of a HPC network (putative hippocampus), which serve as memory labels for the patterns. The connectivity matrix can be constructed in a simple, online, Hebbian way (thus in one-shot) as the outer-product of input and memory patterns. When presented with a noisy version of an input pattern, the memory network robustly retrieves the correct label (and maintains it in the absence of input).

Such a pattern labeler can be used for recognition or familiarity detection, template matching, classification, locality-sensitive hashing and nearest neighbor computations. It could also be used for memory consolidation and the learning of conjunctive representations, filling a gap in general theories of the hippocampal formation by allowing a much smaller hippocampal network to provide and robustly retrieve labels for very high-dimensional input patterns. As illustration, consider recognition memory: our network can rapidly store large numbers of inputs (one-shot learning) and then make robust judgments about familiarity or novelty during testing, compatible with the prodigious recognition memory found in human psychophysics. When presented with a previously learned input pattern, the network dynamics settle to a global energy minimum at the state corresponding to the familiar pattern label. By contrast, when presented with a novel pattern, the network settles to one of the many local minima that populate the spaces between the basins of attraction surrounding the global minima. These local minima are higher-energy states and with their higher number of unsatisfied constraints correspond to a higher level of activation in constraint neurons. A single readout neuron which sums constraint activations can thus signal that the pattern was novel.

**Sensitivity to form of data (R4):** We expect our results will generalize to different neural responses, since the decoding performance is due primarily to the connectivity structure of the network: with expansion, multiple neurons in the constraint layer receive input from only one corrupted input neuron and can identify the error.

**Transition between recovery and failure (R4):** The steep transition is typical of both error-correcting codes and random phenomena in high dimensions, where many events happen with probability that is asymptotically 0 or 1 (zero-one laws). Even when decoding fails, there is information in the location of convergence (which we use in the recognition memory model described above).

[Meta-Review · NeurIPS 2019]

This paper presents a novel form of associative content addressable (ACA) memory systems. The canonical model for ACA memory is the Hopfield network, which can only store approximately N patterns of N bits. The authors use developments from error-correcting codes (ECCs) to implement an ACA that can store e^N, N bit patterns. This is accomplished by using a bipartite expander graph, which is essentially a restricted Boltzmann machine (RBM) wherein the hidden nodes are actually clusters of units that are mutually inhibitory. The authors demonstrate that these networks have dynamics that can engage in error correction similar to ECCs, enabling the storage of exponentially many patterns. The reviewers agreed that this work was theoretically interesting and potentially relevant to neuroscience. The connections made between Hopfield nets, RBMs and ECCs are novel and worthy of presentation at NeurIPS. The reviewers had some concerns, but these were largely addressed in the authors' response. Therefore, the reviewers agreed that this paper should be accepted.